# $J\bar{T}$ - deformed CFTs as non-local CFTs

Monica Guica

*Université Paris-Saclay, CNRS, CEA, Institut de Physique Théorique, 91191 Gif-sur-Yvette, France*

## Abstract

Various holographic set-ups in string theory suggest the existence of non-local, UV complete two-dimensional QFTs that possess Virasoro symmetry, in spite of their non-locality. We argue that $J\bar{T}$- deformed CFTs are the first concrete realisation of such "non-local CFTs", through a detailed analysis of their classical and quantum symmetry algebra.

Classically, the symmetries consist of an infinite set of left-moving conformal and affine $U(1)$ transformations that generate a Witt-Kac-Moody algebra, as well as a set of non-local, field-dependent generalizations of right-moving conformal and affine $U(1)$ transformations, whose algebra depends on the chosen basis. Notably, there exists a basis, denoted as the "flowed" representation, in which the right-moving charge algebra is simply Witt-Kac-Moody. At the quantum level, we provide a concrete prescription for constructing the symmetry generators via a combination of the flow equations they satisfy and the Sugawara construction, and use this to explicitly resolve the ordering ambiguities and the quantum corrections to the generators up to second order in the $J\bar{T}$ coupling parameter. This construction naturally produces the "flowed" generators, whose algebra is Virasoro-Kac-Moody to all orders in the coupling, with the same central extension as that of the undeformed CFT. We use this input to work out the quantum modifications to the "unflowed" generator algebra.

A peculiarity of the Virasoro generators we study is that their zero mode does not equal the Hamiltonian, but is a quadratic function of it; this helps reconcile the Virasoro symmetry with the non-locality of the model. We argue that also $T\bar{T}$ - deformed CFTs posses Virasoro symmetry, and discuss criteria for the existence of such a symmetry in more general non-local QFTs.

# 1. Introduction

Decoupling limits play an important role in the derivation of holographic dualities from string theory, most notably of the AdS/CFT correspondence [1]. However, there exist many decoupling limits in which the spacetime is not asymptotically AdS [2–4]; rather, it corresponds to a non-normalizable deformation of it. A naïve application of holographic reasoning then suggests that such backgrounds are dual to irrelevant deformations of a CFT that nevertheless produce UV complete QFTs. The resulting theory is non-local at the scale set by the irrelevant deformation.

This idea has been realised concretely in a handful of examples, where the irrelevant deformation could be rewritten as a star product deformation of the original CFT Lagrangian [5, 6]. However, in general one does not have access to such explicit additional structures and needs to deal directly with the irrelevant deformation. This is extremely daunting, not least because specifying the irrelevant flow is highly non-trivial. Not even maximal supersymmetry can solve the problem [7].

A particularly interesting class of decoupled backgrounds correspond to irrelevant deformations of a two-dimensional CFT by a $(1, 2)$ operator [8–10]. These backgrounds are related to extremal black holes [10]. As mentioned, it is notoriously hard to specify the flow driven by an irrelevant deformation; however the existence of a decoupling limit suggests that the resulting theory is UV complete. Interestingly, in all these set-ups, asymptotic symmetry group analyses suggest the existence of two sets of the infinite-dimensional Virasoro symmetry, even though the dual QFTs are intrinsically non-local [11].

Symmetries, of course, provide a powerful tool of classification and control over the behaviour of quantum field theories. Conformal symmetries, in particular, allow one to in principle determine all correlation functions of local operators in terms of a set of data encoded in two and three-point functions [12]; these constraints are even more powerful in two dimensions, because the conformal group is infinite-dimensional [13]. It would thus be extremely interesting to establish whether the above conjecturally UV complete QFTs do posses Virasoro symmetry, since it may provide the "additional structure" needed to specify the irrelevant deformations. If it is present, we will denote these theories as "non-local CFTs"[1].

It is however not clear, *a priori*, whether Virasoro symmetries are compatible with non-locality. Since much of the evidence in favour of their existence in non-local theories comes from asymptotic group analyses - computations in the dual classical gravitational picture that suffer from multiple limitations and ambiguities - it is certainly a logical possibility that the Virasoro symmetry they predict is simply an artifact of the classical gravity approximation or, worse, that the gravitational calculations miss subtleties in the construction that ultimately make the charges ill-defined. Thus, in order to be able to explore this interesting possibility, it is crucial to have a well-defined, $QFT$ example of a non-local CFT.

The main claim of this article is that $J\bar{T}$ - deformed CFTs [14], a class of universal irrelevant deformations of two-dimensional CFTs that are in principle well-defined from a quantum-field-theoretical point of view [15], provide the *first concrete example* of a non-local (dipole) CFT. We will establish this fact through a detailed analysis of their classical and quantum symmetries. Since certain properties of these symmetries may appear somewhat peculiar, we start by reminding the reader a few basic facts about the definition of a symmetry.

At the classical level, the simplest way to define a (continuous) symmetry is as a transformation of the fields and, eventually, the coordinates, that leaves the action invariant *off-shell*. One can then employ the algorithmic Noether procedure to construct, for any given Lagrangian, a current and associated charge corresponding to the given symmetry, which are conserved on-shell.

Symmetries can also be defined in the Hamiltonian formalism, usually as corresponding to functions on phase space whose Poisson bracket with the Hamiltonian vanishes. This requirement must be appropriately modified in presence of space-time dependent symmetries such as, for example, boosts, whose Poisson bracket with the Hamiltonian does not vanish; instead

$$\{H, M^{0i}\} = P^i \tag{1.1}$$

i.e., the commutator is a linear combination of the generators of the Poincaré group. In this case, one

---

[1]The example studied here, as well as those of [8–10], posses an additional global $SL(2, \mathbb{R})$ symmetry. Relatedly, they are only 'half non-local'. They have been previously called "dipole CFTs", in analogy with the four-dimensional example [6].

can build a conserved charge by including an explicit time dependence in the generator[2]. The general requirement for function $Q$ on phase space to correspond to a symmetry is then

$$\partial_t Q - \{H, Q\} = 0 \tag{1.2}$$

*off-shell*. Note that the left-hand side of this equation simply equals $\frac{dQ}{dt}$ upon using Hamilton's equations of motion, thus expressing the expected on-shell conservation of the charge. All this does, of course, is to replicate the conserved charges obtained via the Noether procedure, which in the case of space-time symmetries are often explicitly time-dependent, in the Hamiltonian formalism.

The quantum-mechanical definition of symmetries parallels the discussion above, in that the symmetry generators are usually expected to commute with the Hamiltonian. However, in the case of space-time symmetries, this requirement is relaxed to allow e.g. for commutators that are linear combinations of the symmetry generators, by including some explicit time dependence. The generators should then satisfy

$$\left(\frac{\partial L_S}{\partial t}\right)_H + \frac{i}{\hbar}[H, L_H] = 0 \tag{1.3}$$

where the subscripts $S, H$ denote the Scrödinger and, respectively, the Heisenberg picture operator. Upon using Heisenberg's equation of motion, the above condition encodes the time-independence of the Heisenberg picture operator $L_H$, which implies that its expectation value is conserved in any state.

As advertised, the focus of this article is on the symmetries of $J\bar{T}$ - deformed CFTs, a class of solvable [14, 16–18] irrelevant deformations of two-dimensional CFTs by an operator bilinear in a $U(1)$ current and the right-moving stress tensor. $J\bar{T}$ - deformed CFTs correspond to a particular case of the Smirnov-Zamolodchikov deformations [15], which also include the well-studied $T\bar{T}$ deformation [15, 19–22]. Our reasons for concentrating on this particular deformation are two-fold: first, $J\bar{T}$ - deformed CFTs are interesting in their own right as solvable, likely UV complete QFTs with non-standard UV behaviour, which are directly relevant to understanding the holographic description of extremal black holes [10, 23]; second, the structure of the deformed theory is particularly simple, in that the deformation preserves an $SL(2, \mathbb{R})$ "left" subgroup of the original conformal group. The resulting theory is thus local and conformal on the left, while all the non-locality induced by the irrelevant deformation is concentrated on the remaining null right-moving direction - a structure which is significantly simpler and easier to control than, for example, that of $T\bar{T}$.

The extended symmetries of (classical) $J\bar{T}$ - deformed CFTs were first investigated from a field-theoretical perspective in [24]. On the local left-moving side, the global conformal and $U(1)$ symmetries were enhanced to form a Witt-Kac-Moody algebra, as expected. Interestingly, also the non-local right-moving side exhibited an infinite number of symmetries, implemented by a field-dependent generalization of right-moving conformal and affine $U(1)$ transformations. These symmetries were denoted as "pseudo-conformal" in [24]. A subsequent analysis [25] noted that in finite volume, the generators of [24] were inconsistent with charge quantization and proposed to slightly modify them, which resolved the issue.

A peculiarity of the new finite-volume generators is that their Poisson bracket with the Hamiltonian takes the form

$$\{H, Q\} = f(H - P, J_0) Q \tag{1.4}$$

where $f$ is a known, non-trivial function of its arguments, which include the momentum $P$ and a global $U(1)$ charge, $J_0$. Even if this bracket may appear unusual, the charges are conserved off-shell via (1.2).

The modification proposed by [25] does affect the charge algebra, in a way that will be spelled out explicitly in this article. Nevertheless, [25] showed that a particular non-linear combination of the new generators, which highly resembles a "spectral flow by the right-moving energy" and is singled out by satisfying, in the classical limit, the same flow equation as the energy eigenstates, still satisfies a Witt-Kac-Moody algebra. The zero mode of this algebra is however not the Hamiltonian, but a non-linear function of it, which provides perhaps the easiest way to understand the appearance of the non-trivial commutator above. The analysis of [25] was classical, though fully non-linear in $\lambda$.

---

[2]In the case of Galilean boosts, for example, the conservation law of the explicitly time-dependent generator equates the total mass $\times$ the position of the center of mass of the system to the total momentum $\times$ time.

In this article, we investigate the survival of these symmetries at the quantum level. For this, we need to resolve any normal-ordering issues that arise in defining the quantum generators, as well as to determine their algebra, including central extensions and possible quantum corrections to the structure constants, which are determined by their classical Poisson bracket counterparts only up to leading order in $\hbar$. As in the classical case, a peculiarity of the quantum generators we study is that their commutator with the Hamiltonian takes the form (1.4). Even though this is not linear in the generators, as we are accustomed to, it is trivial to give them an explicit time dependence, so that they satisfy the conservation requirement (1.3). The only difference with respect to usual spacetime-dependent symmetries, such as boosts, is that the time-dependent multiplicative factors are now operator-valued, whereas before they were c-numbers. In the classical limit, these time-dependent generators reduce to the classical conserved charges, which are known explicitly at full non-linear level in $\lambda$.

This paper is organised as follows. In section 2, we review the classical symmetry generators in compact space, following [24, 25], and provide several different useful ways to write their algebra. In section 3, we discuss possible definitions of the quantum symmetry generators and use them to explicitly construct the generators up to second order in the $J\bar{T}$ coupling parameter. We also work out the algebra of a set of generators denoted as the "unflowed" ones to all orders in $\hbar$, finding non-trivial modifications with respect to the semiclassical result, and discuss the organisation of states into representations of the symmetry algebra. In section 4, we extract a set of general criteria for the existence of Virasoro symmetry in a non-local, UV complete QFT, and use them to argue that more general theories, such as $T\bar{T}$ - deformed CFTs, also posses Virasoro symmetry. We conclude in 5 with a short discussion. In appendix A, we gather a few useful formulae.

## 2. Classical conserved charges and their algebra

In this section, we review and clarify the results of [24] and [25] on the infinite classical symmetries of $J\bar{T}$ - deformed CFTs. In 2.1, we start with a brief review of $J\bar{T}$ - deformed CFTs, we then list the necessary conditions for having a certain infinite set of symmetries and we display the explicit expressions for the associated conserved charges. In section 2.2, we present the Poisson bracket algebra of these charges and the flow equations they satisfy; in 2.3, we display an equivalent way of encoding the same information, which will be useful in the subsequent quantum analysis. We work exclusively in the Hamiltonian formalism, which is best suited for studying the algebra of the conserved charges.

## 2.1. Infinite classical symmetries of $J\bar{T}$ - deformed CFTs on a cylinder

In the Hamiltonian formalism, $J\bar{T}$ - deformed CFTs are defined via the flow equation

$$\partial_\lambda \mathcal{H} = -\epsilon^{\alpha\beta} J_\alpha T_{\beta V} \tag{2.1}$$

where $\mathcal{H}$ is the Hamiltonian density, $J$ is a $U(1)$ current, $\lambda$ is the deformation parameter, and $T_{\beta V}$ is the generator of translations in the null direction $V$. The theory is defined on the plane or the cylinder (of circumference $R$), with coordinates $t, \sigma$. We will also employ the null coordinates $U, V = \sigma \pm t$.

The components of the stress tensor are determined from the Hamiltonian in the usual way. If we model the $U(1)$ current as a topologically conserved current for a boson $\phi$, which should be generally possible, and assume that the undeformed theory at $\lambda = 0$ is a CFT, then the above equation can be solved exactly, with solution[3] [24]

$$\mathcal{H}_R = \frac{2}{\lambda^2 k}\left(1 - \lambda \mathcal{J}_+ - \sqrt{(1 - \lambda \mathcal{J}_+)^2 - \lambda^2 k \mathcal{H}_R^{(0)}}\right) \tag{2.2}$$

In the above, $\mathcal{H}_{L,R} = \frac{1}{2}(\mathcal{H} \pm \mathcal{P})$ are the left- and, respectively, right-moving Hamiltonian densities, $\mathcal{H}^{(0)}$ is the Hamiltonian of the undeformed CFT and $\mathcal{J}_\pm = \sqrt{k}(\pi \pm \phi')/2$, where $\phi' = \partial_\sigma \phi$ and $\pi$ is the conjugate

---

[3]The derivation of [24] was for level $k = 1$. The result for general $k$ can be simply obtained via the rescaling $\mathcal{J}_+ \to \mathcal{J}_+/\sqrt{k}$ and $\lambda \to \lambda\sqrt{k}$.

momentum to $\phi$. This is a universal formula for the classical $J\bar{T}$ - deformed Hamiltonian density to all orders in $\lambda$ that only depends on the universal conserved currents of the seed theory.

The Poisson brackets of the Hamiltonian currents $\mathcal{H}_{L,R}$ follow from the Poisson brackets of their undeformed counterparts, see [24] or [25] for the explicit expressions. In particular, their commutation relations with the total Hamiltonian $H = \int d\sigma(\mathcal{H}_L + \mathcal{H}_R)$ are given by

$$\{H, \mathcal{H}_L\} = -\partial_\sigma \mathcal{H}_L \, , \qquad \{H, \mathcal{H}_R\} = \partial_\sigma \left( \mathcal{H}_R \frac{1 + \lambda \mathcal{J}_+ + \frac{\lambda^2 k}{2} \mathcal{H}_R}{1 - \lambda \mathcal{J}_+ - \frac{\lambda^2 k}{2} \mathcal{H}_R} \right) \tag{2.3}$$

The first of these equations implies the holomorphic conservation of $\mathcal{H}_L$ on-shell. There is also a holomorphic $U(1)$ current, with

$$K_t = K_\sigma = \mathcal{J}_+ + \frac{\lambda k}{2} \mathcal{H}_R \, , \qquad \{H, K_t\} = -\partial_\sigma K_t \tag{2.4}$$

We will also consider a right-moving $U(1)$ current $\bar{K}_\alpha$, defined in [24], whose time component is given by

$$\bar{K}_t = \mathcal{J}_- + \frac{\lambda k}{2} \mathcal{H}_R \, , \qquad \{H, \bar{K}_t\} = \partial_\sigma \left( \bar{K}_t \frac{1 + \lambda \mathcal{J}_+ + \frac{\lambda^2 k}{2} \mathcal{H}_R}{1 - \lambda \mathcal{J}_+ - \frac{\lambda^2 k}{2} \mathcal{H}_R} \right) \tag{2.5}$$

As reviewed in the introduction, symmetries correspond to functions on phase space that satisfy

$$\partial_t Q - \{H, Q\} = 0 \tag{2.6}$$

off-shell. It is easy to see that the quantities

$$Q_f = \int d\sigma f(U) \mathcal{H}_L \, , \qquad P_\eta = \int d\sigma \eta(U) \left( \mathcal{J}_+ + \frac{\lambda k}{2} \mathcal{H}_R \right) \tag{2.7}$$

labeled by the *arbitrary* functions $f, \eta$ of the null coordinate $U = \sigma + t$, do satisfy the requirement (2.6). These represent the expected left conformal and affine $U(1)$ symmetries of $J\bar{T}$ - deformed CFTs, as can be checked by computing the change in $\phi$ induced by these transformations. Their Poisson bracket algebra is the Witt-Kac-Moody algebra. On a cylinder, which is the main setup of this article, we will work in the Fourier basis $f_n = e^{inU/R}$, and denote the corresponding conserved charges as $Q_n, P_n$. The zero mode of $P_\eta$ will be denoted as $Q_K = J_0 + \lambda k E_R/2$.

These Witt-Kac-Moody symmetries were to be expected, based on the locality and left-moving conformal invariance of the model. More non-trivially, [24] found that the action was also invariant under a set of field-dependent coordinate transformations, of the form

$$V \rightarrow V - \epsilon \bar{f}(v) \tag{2.8}$$

where $v$ is a possibly field-dependent coordinate. The generator of this transformation is

$$\bar{Q}_{\bar{f}} = \int d\sigma \bar{f}(v) \mathcal{H}_R \tag{2.9}$$

as expected from the fact that $\mathcal{H}_R = -T_{tV}$. The condition (2.6) that these transformations correspond to a symmetry for any function $\bar{f}$ can be written as a condition on $v$, and reads

$$\frac{\partial_t v - \{H, v\}}{\partial_\sigma v} = -\frac{1 + \lambda \mathcal{J}_+ + \frac{\lambda^2 k}{2} \mathcal{H}_R}{1 - \lambda \mathcal{J}_+ - \frac{\lambda^2 k}{2} \mathcal{H}_R} = -\frac{1 - \lambda \{H, \phi\}}{1 - \lambda \phi'} \tag{2.10}$$

where for the first equation we used (2.3), while the second is simply an identity[4]. Thus, if $v$ satisfies the equation above, we obtain an infinite set of classical symmetries, since the function $\bar{f}$ is arbitrary. We call these symmetries pseudoconformal, since they are implemented by a field-dependent modification of right-moving conformal transformations.

---

[4]This identity follows from the Poisson bracket $\{H, \phi\} = -\frac{2\mathcal{J}_- + \lambda k \mathcal{H}_R}{1 - \lambda \mathcal{J}_+ - \lambda^2 k \mathcal{H}_R/2} - \mathcal{J}_+ + \mathcal{J}_-$.

One can furthermore show that the quantities

$$\bar{P}_{\bar{\eta}} = \int d\sigma \bar{\eta}(v) \left( \mathcal{J}_- + \frac{\lambda k}{2} \mathcal{H}_R \right) \tag{2.11}$$

are also conserved, provided $v$ satisfies the constraint (2.10). These are conserved charges associated to a set of field-dependent affine transformations that generalize the right-moving $U(1)$. See [24] for more details. The zero mode of $\bar{P}_{\bar{\eta}}$ will be denoted as $\bar{Q}_{\bar{K}} = \bar{J}_0 + \lambda k E_R/2$. Note that we have dropped the superscript 'KM' with respect to the notation in [24].

The simplest solution to the equation (2.10) above is

$$v = \sigma - t - \lambda\phi \tag{2.12}$$

The Poisson brackets of the corresponding charges (2.9), (2.11) yield another copy of the Witt-Kac-Moody algebra, which entirely commutes with the left-moving one [24].

To study states and representations, it is natural to consider the charge algebra of $J\bar{T}$ - deformed CFTs on a cylinder, taken to have circumference $R$. Charge conservation then requires that $\bar{f}$ be periodic, which amounts to considering a basis of periodic functions $\bar{f}_n = e^{-inv/R_v}$, where $R_v$ is the field-dependent radius of $v$. From the definition (2.12), we have

$$R_v = R - \lambda w \tag{2.13}$$

where $w$ is the winding of the field $\phi$ around the circle. This introduces explicit factors of $R_v$ in the algebra of Fourier modes, as discussed in [24]. It is the rescaled generators $R_v \bar{Q}_n$ that satisfy the usual Witt algebra at the level of Fourier modes.

However, in the case of $J\bar{T}$ - deformed CFTs on a cylinder, it was pointed out in [25] that the Poisson brackets of the right-moving generators $\bar{Q}_{\bar{f}}, \bar{P}_{\bar{\eta}}$ above with the global $U(1)$ charge and the momentum are inconsistent with the quantization of these charges in the deformed theory. The problem roughly stems from the fact that the field-dependent coordinate $v$ above contains the zero mode $\phi_0 = \frac{1}{R} \int d\sigma\phi$ of the field $\phi$, whose non-trivial commutation relations with the global $U(1)$ charges

$$J_0 \equiv \int d\sigma \, \mathcal{J}_+ \, , \qquad \bar{J}_0 \equiv \int d\sigma \, \mathcal{J}_- \tag{2.14}$$

lead to the above inconsistency.

This problem can be remedied by noting that the equation (2.10) only determines $v$ up to a constant term. In [25], it was proposed to modify $v$ as

$$v \to v_{imp} = \sigma - t - \lambda\phi + \frac{\lambda\widetilde{\phi}_0 R_v}{R - \lambda Q_K} - \frac{\lambda(Q_K R_v + \bar{Q}_{\bar{K}}R)}{R(R - \lambda Q_K)}t = v + \frac{\lambda(\widetilde{\phi}_0 - \alpha\,t)R_v}{R - \lambda Q_K} \tag{2.15}$$

where $\widetilde{\phi}_0$ is the spectral flow generator in the full $J\bar{T}$ - deformed CFT, which takes the form

$$\widetilde{\phi}_0 = \phi_0 - \frac{\lambda}{R_v} \int d\sigma \left( \mathcal{J}_- + \frac{\lambda k}{2} \mathcal{H}_R \right) \hat{\phi} \tag{2.16}$$

where $\hat{\phi} \equiv \phi - w\sigma/R$ is the scalar with its winding mode removed, which is thus single-valued on the circle. As shown in [25] , $\widetilde{\phi}_0$ has remarkably simple Poisson brackets with the symmetry generators (2.7), (2.9) and (2.11), namely

$$\{\widetilde{\phi}_0, Q_f\} = \frac{P_f}{R} \, , \qquad \{\widetilde{\phi}_0, P_\eta\} = \frac{k}{2R} \int d\sigma\eta \, , \qquad \{\widetilde{\phi}_0, \bar{Q}_{\bar{f}}\} = \frac{\bar{P}_{\bar{f}}}{R_v} \, , \qquad \{\widetilde{\phi}_0, \bar{P}_{\bar{\eta}}\} = \frac{k}{2R_v} \int d\sigma\bar{\eta}(1 - \lambda\phi') \tag{2.17}$$

Remarkably, the new field-dependent coordinate $v_{imp}$ still satisfies (2.10), since the redefiniton (2.15) does not affect either the space, nor the time derivative of $v$. The former is obvious; the latter follows from

$$\partial_t \Delta v - \{H, \Delta v\} = -\frac{\lambda(Q_K R_v + \bar{Q}_{\bar{K}}R)}{R(R - \lambda Q_K)} - \frac{\lambda R_v}{R - \lambda Q_K}\{H, \widetilde{\phi}_0\} = 0 \tag{2.18}$$

derived using (2.17), where $\Delta v = v_{imp} - v$. Even if the change $v \to v_{imp}$ looks rather innocuous in the Hamiltonian formalism, its impact on the transformation of the fields such as $\phi$ is quite non-trivial, due to the additional commutator $\{v_{imp}, \phi\}$. This introduces, in addition to the field-dependent coordinate shift by $\bar{f}(v_{imp})$, a fully non-local term in the transformation law for $\phi$, whose physical interpretation in the Lagrangian formalism is not very clear. Leaving aside the highly non-local nature of the transformation, the associated charge is conserved, as per the general arguments we gave.

Writing

$$\hat{\phi} = \phi_0 + \hat{\phi}_{nzm} \tag{2.19}$$

one can show that $\phi_0$ entirely drops out form the expression for $v_{imp}$; concretely

$$v_{imp} = \frac{R_v}{R}\sigma - \lambda\hat{\phi}_{nzm} - \frac{\lambda^2}{R - \lambda Q_k}\int d\sigma\left(\mathcal{J}_- + \frac{\lambda k}{2}\mathcal{H}_R\right)\hat{\phi}_{nzm} - \frac{R_v(R + \lambda Q_K)}{R(R - \lambda Q_K)}t \tag{2.20}$$

and thus the commutators with the global $U(1)$ charges will now vanish.

To summarize, we find that the conserved charges whose action is well-defined on the phase space of the $J\bar{T}$ - deformed CFT on the cylinder are still given by (2.9), but with $v$ replaced by $v_{imp}$. We will denote these generators by barred capital calligraphic letters, $\bar{\mathcal{Q}}, \bar{\mathcal{P}}$, to differentiate them from $\bar{Q}, \bar{P}$, which are not well-defined on the cylinder. Explicitly,

$$\bar{\mathcal{Q}}_{\bar{f}} = \int d\sigma \bar{f}(v_{imp})\mathcal{H}_R \,, \qquad \bar{\mathcal{P}}_{\bar{\eta}} = \int d\sigma \bar{\eta}(v_{imp})\left(\mathcal{J}_- + \frac{\lambda k}{2}\mathcal{H}_R\right) \tag{2.21}$$

These charges are: i) conserved and ii) consistent with semiclassical quantization. The relationship between the two sets of charges is particularly simple in the Fourier basis $\bar{f}_n = e^{-inv_{imp}/R_v}$, and reads

$$\bar{\mathcal{Q}}_n = \bar{Q}_n\, e^{-\frac{in\lambda(\tilde{\phi}_0 - \alpha t)}{R - \lambda Q_K}} \,, \qquad \bar{\mathcal{P}}_n = \bar{P}_n\, e^{-\frac{in\lambda(\tilde{\phi}_0 - \alpha t)}{R - \lambda Q_K}} \tag{2.22}$$

Note that $\alpha$, defined in (2.15), only depends on $Q_K$ and $\bar{Q}_{\bar{K}}$, and thus commutes with all the $\bar{Q}_m, \bar{P}_n$.

## 2.2. Poisson bracket algebra and flow of the symmetry generators

The Poisson brackets of the left-moving generators were worked out in [24], and read

$$\{Q_m, Q_n\} = -\frac{i(m-n)}{R}Q_{m+n} \,, \qquad \{Q_m, P_n\} = \frac{in}{R}P_{m+n} \,, \qquad \{P_m, P_n\} = -\frac{imk}{2}\delta_{m+n} \tag{2.23}$$

The Poisson brackets of the right-moving generators can be derived either from their explicit expressions (2.21), (2.2) in terms of the undeformed currents or, equivalently, from the known algebra of the $\bar{Q}_m, \bar{P}_m$ and $\tilde{\phi}_0$, using (2.22). We obtain the following non-linear modification of the Witt-Kac-Moody algebra

$$\{\bar{\mathcal{P}}_m, \bar{\mathcal{P}}_n\} = -\frac{imk}{2}\delta_{m+n} - \frac{im\lambda k}{2(R - \lambda Q_K)}\delta_{n,0}\bar{\mathcal{P}}_m + \frac{in\lambda k}{2(R - \lambda Q_K)}\delta_{m,0}\bar{\mathcal{P}}_n$$

$$\{\bar{\mathcal{Q}}_m, \bar{\mathcal{P}}_n\} = \frac{in}{R_v}\bar{\mathcal{P}}_{m+n} - \frac{im\lambda k}{2(R - \lambda Q_K)}\bar{\mathcal{Q}}_m\delta_{n,0} + \frac{in\lambda}{R_v(R - \lambda Q_K)}\bar{\mathcal{P}}_m\bar{\mathcal{P}}_n$$

$$\{\bar{\mathcal{Q}}_m, \bar{\mathcal{Q}}_n\} = -\frac{i(m-n)}{R_v}\bar{\mathcal{Q}}_{m+n} - \frac{im\lambda}{R_v(R - \lambda Q_K)}\bar{\mathcal{Q}}_m\bar{\mathcal{P}}_n + \frac{in\lambda}{R_v(R - \lambda Q_K)}\bar{\mathcal{P}}_m\bar{\mathcal{Q}}_n \tag{2.24}$$

where we have reinstated the factors of the level in an appropriate way. The commutation relations with the left-moving charges read

$$\{Q_m, \bar{\mathcal{Q}}_n\} = \frac{in\lambda}{R(R - \lambda Q_K)}\bar{\mathcal{Q}}_n P_m \,, \qquad \{Q_m, \bar{\mathcal{P}}_n\} = \frac{in\lambda}{R(R - \lambda Q_K)}\bar{\mathcal{P}}_n P_m$$

$$\{P_m, \bar{\mathcal{Q}}_n\} = \frac{in\lambda k}{2(R - \lambda Q_K)}\bar{\mathcal{Q}}_n\delta_{m,0} \,, \qquad \{P_m, \bar{\mathcal{P}}_n\} = \frac{in\lambda k}{2(R - \lambda Q_K)}\bar{\mathcal{P}}_n\delta_{m,0} \tag{2.25}$$

yielding again non-linear terms on the right-hand side.

We would now like to connect this to the results of [25], who showed that particular non-linear combinations of the above generators satisfy a Witt-Kac-Moody algebra. Concretely, these combinations are

$$\widetilde{P}_n \equiv P_n - \frac{\lambda k E_R}{2}\delta_{n,0} \quad \text{and} \quad \widetilde{Q}_n \equiv Q_n - \frac{\lambda E_R}{R}P_n + \frac{\lambda^2 k E_R^2}{4R}\delta_{n,0}$$

$$\widetilde{\bar{\mathcal{P}}}_n \equiv \bar{\mathcal{P}}_n - \frac{\lambda k E_R}{2}\delta_{n,0} \quad \text{and} \quad \widetilde{\bar{\mathcal{Q}}}_n \equiv \frac{R_v}{R}\bar{\mathcal{Q}}_n - \frac{\lambda E_R}{R}\bar{\mathcal{P}}_n + \frac{\lambda^2 k E_R^2}{4R}\delta_{n,0} \tag{2.26}$$

where $E_R = \int d\sigma \mathcal{H}_R$ is the right-moving energy, alternativey denoted as $\bar{\mathcal{Q}}_0$. Its Poisson brackets are given by

$$\{E_R, \bar{\mathcal{P}}_n\} = \frac{in}{R - \lambda Q_K}\bar{\mathcal{P}}_n , \qquad \{E_R, \bar{\mathcal{Q}}_n\} = \frac{in}{R - \lambda Q_K}\bar{\mathcal{Q}}_n \tag{2.27}$$

Using these, one can easily show that the commutation relations of (2.26), which we will denote as the "flowed" generators, consist of two commuting copies of the Witt-Kac-Moody algebra.

The relation (2.26) between the two sets of generators is reminiscent of spectral flow (see the next subsection for a review), though the parameter of the flow is not a number, but the right-moving energy, $E_R$. Due to this fact, the transformation changes the algebra[5]. Note that at the level of the currents, the spectral flow operation simply replaces the left-moving zero mode $Q_K = J_0 + \lambda E_R k/2$ by $J_0$ and, similarly, $\bar{Q}_{\bar{K}}$ by $\bar{J}_0$ on the right-moving side.

As detailed in [25], the flowed and unflowed conserved charges satisfy slightly different flow equations with respect to the parameter $\lambda$ of the deformation. At $t = 0$, we have

$$\mathcal{D}_\lambda Q_m \equiv \partial_\lambda Q_m - i\{\mathcal{O}_{cls}, Q_m\} = 0 \tag{2.28}$$

where $\mathcal{O}_{cls}$ is the sum of two parts, $\hat{\mathcal{O}}_{cls}$ and $\Delta\mathcal{O}_{cls}$, given by[6]

$$\hat{\mathcal{O}}_{cls} = i\int d\sigma d\tilde{\sigma} G(\sigma - \tilde{\sigma})\phi'(\sigma)\mathcal{H}_R(\tilde{\sigma}) , \qquad \Delta\mathcal{O}_{cls} = i(w\chi_0 - \phi_0 E_R) \tag{2.29}$$

In the above, $G$ is the Green's function on the cylinder and $\chi_0$ is the zero mode of a field $\chi$ defined though $\partial_\sigma \chi = \mathcal{H}_R$. The above flow equation is also satisfied by $P_m$, $R_v\bar{\mathcal{Q}}_m$ and $\bar{P}_m$. As explained, of these, $\bar{Q}_m, \bar{P}_m$ will not lift to well-defined quantum symmetry generators on the cylinder, but only $\bar{\mathcal{Q}}_m, \bar{\mathcal{P}}_m$ will. The flow equations for $\bar{\mathcal{Q}}_m, \bar{\mathcal{P}}_m$, given in (2.22), follow from the one above and the fact that $\mathcal{D}_\lambda \phi_0 = 0$ at $t = 0$, so only the $\lambda$ derivatives contribute. Concretely,

$$\mathcal{D}_\lambda \bar{\mathcal{P}}_m = -\frac{im\widetilde{\phi}_0 R}{(R - \lambda Q_K)^2}\bar{\mathcal{P}}_m , \qquad \mathcal{D}_\lambda(R_v\bar{\mathcal{Q}}_m) = -\frac{im\widetilde{\phi}_0 R}{(R - \lambda Q_K)^2}R_v\bar{\mathcal{Q}}_m \tag{2.30}$$

One can check that these flow equations are consistent with the commutation relations (2.24). Note that the explicit $\lambda$ dependence of the algebra of these generators is directly related to the fact that the flow equations they obey are not homogenous. On the other hand, $\widetilde{Q}_m, \widetilde{P}_m, \widetilde{\bar{\mathcal{P}}}_m$ and $R_v\widetilde{\bar{\mathcal{Q}}}_m$ satisfy

$$\tilde{\mathcal{D}}_\lambda \widetilde{Q}_m \equiv \partial_\lambda \widetilde{Q}_m - i\left\{\tilde{\mathcal{O}}_{cls}, \widetilde{Q}_m\right\} = 0 \tag{2.31}$$

where $\tilde{\mathcal{O}}_{cls} = \hat{\mathcal{O}}_{cls} + \Delta\tilde{\mathcal{O}}_{cls}$, with $\hat{\mathcal{O}}_{cls}$ as given above and

$$\Delta\tilde{\mathcal{O}}_{cls} \equiv i\left[w(\chi_0 - \widetilde{\chi}_0) - \phi_0 E_R + \frac{\widetilde{\phi}_0 E_R R}{R - \lambda Q_K}\right] \tag{2.32}$$

---

[5]It may be more appropriate to rather call the relation between $\bar{Q}, \bar{P}$ and $\widetilde{\bar{\mathcal{Q}}}, \widetilde{\bar{\mathcal{P}}}$ spectral flow, i.e. conjugation by the spectral flow operator, as discussed in [25]. With this definition, the algebra is naturally unchanged. In either case, the terminology "flowed generators" will apply to the ones defined in (2.26).

[6]Note we have changed the definition of $\hat{\mathcal{O}}, \Delta\mathcal{O}$ by a factor of $i$ with respect to [25], in order to avoid dealing with explicit factors of $i$ at the quantum level. We have also added the subscript 'cls' in order to differentiate these classical functions on phase space from the corresponding operators we will discuss in the next section.

which is the same flow equation (in the classical limit) as that satisfied by the energy eigenstates[7]. Since the flow equation is homogenous and the algebra at $\lambda = 0$ is Witt-Kac-Moody, it follows that it will continue to be so at nonzero $\lambda$. Our explicit check above confirms this more indirect argument.

A nice feature of the flow equation (2.31) is that it does not contain any $\phi_0, \chi_0$. Explicitly,

$$\Delta\tilde{\mathcal{O}}_{cls} = -\frac{iw\lambda}{R_v}\int d\sigma \mathcal{H}_R\hat{\phi}_{nzm} - \frac{i\lambda E_R R}{R_v(R-\lambda Q_K)}\int d\sigma(\mathcal{J}_- + \frac{\lambda}{2}\mathcal{H}_R)\hat{\phi}_{nzm} \tag{2.33}$$

where $\hat{\phi}_{nzm}$ is the scalar field with its winding and zero mode removed (2.19). One can also work out the flow of the right-moving generators with respect to $\tilde{\mathcal{D}}_\lambda$, finding

$$\tilde{\mathcal{D}}_\lambda\bar{\mathcal{P}}_m = \frac{kRE_R}{2(R-\lambda Q_K)}\delta_{m,0} \ , \qquad \tilde{\mathcal{D}}_\lambda(R_v\bar{\mathcal{Q}}_m) = \frac{RE_R}{R-\lambda Q_K}\bar{\mathcal{P}}_m \tag{2.34}$$

These equations are still non-homogenous, though perhaps simpler than (2.30). Note that the second flow equation will suffer from ordering ambiguities at the quantum level.

To summarize, we showed that classical $J\bar{T}$ - deformed CFTs on a cylinder posses two sets of symmetry generators of interest. The ones that are the closest to the original conformal generators, which are given by (2.7), (2.21), have the desireable property that their zero mode coincides with the left- and, respectively, right-moving energy; however, their algebra is not simple. One can also consider the spectrally flowed generators (2.26), whose advantage is that they satisfy the well-studied Witt-Kac-Moody algebra. Their zero modes, however, are non-linearly related to the left-/right-moving energy. At the quantum level, this feature is responsible for resolving the naïve tension one may perceive between having a Virasoro symmetry algebra and the non-trivial $J\bar{T}$ - deformed spectrum [25]. Both sets of generators are rather different from their counterparts in the original CFT, as they both have a non-trivial dependence on $\lambda$.

## 2.3. Spectral flow-invariant generators and a Sugawara-type construction

To better understand the relation between the two sets of generators and, especially, to prepare the ground for their generalisation to the quantum case, in this section we would like to remind a few facts about (quantum) CFTs that posses a $U(1)$ current algebra in addition to Virasoro.

Denoting the Virasoro generators by $L_m$ and the Kac-Moody ones by $J_m$, their algebra is

$$[L_m, L_n] = (m-n)L_{m+n} + \frac{c}{12}m(m^2-1)\delta_{m+n} \ , \qquad [L_m, J_n] = -nJ_{m+n} \ , \qquad [J_m, J_n] = \frac{km}{2}\delta_{m+n} \tag{2.35}$$

The modes $L_m$ of the stress tensor can be written as a Sugawara contribution from the modes of the current, and a remainder $\hat{L}_m$

$$L_m = \hat{L}_m + \frac{1}{k}\sum_\ell : J_{-\ell}J_{m+\ell} : \tag{2.36}$$

where the ':' denote creation-annihilation normal ordering, herein defined as

$$: J_{-\ell}J_{m+\ell} : \equiv \begin{cases} J_{-\ell}J_{m+\ell} & if \quad \ell > 0 \\ J_{m+\ell}J_{-\ell} & if \quad \ell \leq 0 \end{cases} \tag{2.37}$$

Of course, other choices of normal ordering are possible, but they are irrelevant for this discussion. The algebra of $\hat{L}_m$ and $J_m$ is

$$[\hat{L}_m, \hat{L}_n] = (m-n)\hat{L}_{m+n} + \frac{c-1}{12}m(m^2-1)\delta_{m+n} \ , \qquad [\hat{L}_m, J_n] = 0 \ , \qquad [J_m, J_n] = \frac{km}{2}\delta_{m+n} \tag{2.38}$$

The shift of the central charge by one is obtained through a careful analysis of the normal ordering in (2.36). This term is not important in this section, but will be in the quantum analysis of the later ones.

---

[7]In [25], $\Delta\tilde{\mathcal{O}}$ was denoted as $\Delta\mathcal{O} - D$. The combination $\tilde{\chi}_0 \equiv \chi_0 + \frac{\lambda}{R_v}\int d\sigma\mathcal{H}_R\hat{\phi}$, and we have rescaled $\phi_0, \chi_0$ by a factor of $R$.

The algebra (2.35) admits a "spectral flow" outer automorphism labeled by a parameter $\alpha$, under which [26]

$$J_m \to J_m + \frac{\alpha k}{2} \delta_{m,0} , \qquad L_m \to L_m + \alpha J_m + \frac{k\alpha^2}{4} \delta_{m,0} \qquad (2.39)$$

The change in the modes of the stress tensor, $L_m$, under this transformation comes entirely from the part of the stress tensor that is associated to the current via the Sugawara construction, i.e. the transformation above leaves $\hat{L}_m$ invariant. We will thus call the $\hat{L}_m$ the spectral-flow-invariant generators.

The discussion above is fully quantum-mechanical. In this section, we will only use the classical limit of this construction, which consists in dropping the central extension of the Virasoro algebra (but not that of the current algebra, which can be seen at the level of Poisson brackets), as well as the normal ordering in the Sugawara stress tensor.

We would now like to understand whether the conserved (pseudo)conformal charges in $J\bar{T}$ - deformed CFTs can be similarly split into a spectral-flow-invariant contribution and a Sugawara one, associated with the $U(1)$ current. To see this, it is best to start by looking at the single boson case where, if this expectation is true, the entire stress tensor should be given by the Sugawara contribution.

For a free boson ($k = 1$), the Hamiltonian and momentum densities are

$$\mathcal{H}^{(0)} = \frac{\pi^2 + \phi'^2}{2} , \quad \mathcal{P} = \pi\phi' \qquad \Rightarrow \qquad \mathcal{H}^{(0)}_{L,R} = \frac{(\pi \pm \phi')^2}{4} = \mathcal{J}_{\pm}^2 \qquad (2.40)$$

Plugging this into the general formula[8] (2.2) for the deformed Hamiltonian, we can show that

$$\mathcal{H}_L = \left( \mathcal{J}_+ + \frac{\lambda\mathcal{H}_R}{2} \right)^2 = K_t^2 , \qquad \mathcal{H}_R = \frac{\left( \mathcal{J}_- + \frac{\lambda\mathcal{H}_R}{2} \right)^2}{1 - \lambda\phi'} = \frac{\bar{K}_t^2}{v'} \qquad (2.42)$$

where the left- and respectively, right-moving currents $K_t$, $\bar{K}_t$ were defined in (2.4), (2.5) and $v' = v'_{imp}$. Since $\mathcal{H}_L$ is the square of the holomorphic current $K_t$, it immediately follows that the left-moving conformal charges are precisely of the Sugawara type, when one expresses this relation in terms of Fourier modes. For the right-moving sector, a simple manipulation yields

$$\bar{\mathcal{Q}}_n^{(fb)} = \int d\sigma e^{-\frac{in v_{imp}}{R_v}} \frac{\bar{K}_t^2}{v'} = \int d\sigma d\tilde{\sigma} e^{-in\frac{v_{imp}}{R_v}} \bar{K}_t(\sigma) \frac{\delta(\sigma - \tilde{\sigma})}{v'} \bar{K}_t(\tilde{\sigma}) = \frac{1}{R_v} \sum_m \bar{\mathcal{P}}_m \bar{\mathcal{P}}_{n-m} \qquad (2.43)$$

where we rewrote the $\delta$ function as

$$\frac{\delta(\sigma - \tilde{\sigma})}{v'} = \delta(v_{imp} - \tilde{v}_{imp}) = \frac{1}{R_v} \sum_m e^{\frac{im(v_{imp} - \tilde{v}_{imp})}{R_v}} \qquad (2.44)$$

and then performed the $\sigma$ and $\tilde{\sigma}$ integrals to obtain the modes of the right-moving current. Thus, it is indeed true that the right-moving pseudoconformal charges are given by a Sugawara-type expression in terms of the right-moving $U(1)$ current, even though the stress tensor is not itself the square of the current in position space.

It is easy to check that in the case of the free boson, the charges $\bar{\mathcal{Q}}_n^{(fb)}$ given by the Sugawara expression above satisfy the algebra (2.24). More generally, we would write

$$Q_n = \hat{Q}_n + \frac{1}{kR} \sum_m P_m P_{n-m} , \qquad \bar{Q}_n = \hat{\bar{Q}}_n + \frac{1}{kR_v} \sum_m \bar{\mathcal{P}}_m \bar{\mathcal{P}}_{n-m} \qquad (2.45)$$

In terms of the integrated currents, the expression for $\hat{Q}_m$ is given by

$$\hat{Q}_m = \int d\sigma e^{\frac{imU}{R}} \left( \mathcal{H}_L - \frac{1}{k}(\mathcal{J}_+ + \frac{\lambda k}{2}\mathcal{H}_R)^2 \right) = \int d\sigma e^{\frac{imU}{R}} \left( \mathcal{H}_L^{(0)} - \frac{\mathcal{J}_+^2}{k} \right) \qquad (2.46)$$

---

[8]A useful rewriting of this formula is

$$\mathcal{H}_R \left( 1 - \lambda\mathcal{J}_+ - \frac{\lambda^2 k}{4}\mathcal{H}_R \right) = \mathcal{H}_R^{(0)} \qquad (2.41)$$

where we used (2.41) and the fact that $\mathcal{H}_L - \mathcal{H}_R = \mathcal{H}_L^{(0)} - \mathcal{H}_R^{(0)} = \mathcal{P}$. We conclude that the spectral flow invariant left generators in the deformed theory are identical to the undeformed ones, i.e.

$$\hat{Q}_m(\lambda) = \hat{Q}_m^{(0)} \tag{2.47}$$

For the right-movers, we find

$$\hat{\bar{\mathcal{Q}}}_m = \int d\sigma e^{-\frac{imv_{imp}}{R_v}} \left( \mathcal{H}_R - \frac{(\mathcal{J}_- + \frac{\lambda k}{2}\mathcal{H}_R)^2}{k(1-\lambda\phi')} \right) = \int d\sigma e^{-\frac{imv_{imp}}{R_v}} \frac{\mathcal{H}_R^{(0)} - \mathcal{J}_-^2/k}{1-\lambda\phi'} \tag{2.48}$$

What is nice about this expression is that in the last term, the numerator can be expanded in Fourier modes using the original CFT expansion. We obtain

$$\hat{\bar{\mathcal{Q}}}_m = \frac{1}{R_v} e^{\frac{imt(R+\lambda Q_K)}{R(R-\lambda Q_K)} + \frac{im\lambda^2}{R_v(R-\lambda Q_K)} \int d\sigma(\mathcal{J}_-+\frac{\lambda}{2}\mathcal{H}_R)\hat{\phi}_{nzm}} \sum_n \hat{\bar{\mathcal{Q}}}_n^{(0)} e^{-\frac{int}{R}} \int \frac{d\sigma}{1-\frac{\lambda\hat{\phi}'R}{R_v}} e^{\frac{i(n-m)\sigma}{R} + im\lambda\frac{\hat{\phi}_{nzm}}{R_v}} \tag{2.49}$$

which is a significantly simpler expression than that for $\bar{\mathcal{Q}}_m$. The Poisson brackets of $\hat{\bar{\mathcal{Q}}}_m$ can be derived either from the explicit expression above, or by subtracting the Sugawara contribution from the $\bar{\mathcal{Q}}_m$ commutators, and are given by

$$\{\hat{\bar{\mathcal{Q}}}_m, \hat{\bar{\mathcal{Q}}}_n\} = -\frac{i}{R_v}(m-n)\hat{\bar{\mathcal{Q}}}_{m+n} , \qquad \{\hat{\bar{\mathcal{Q}}}_m, \bar{\mathcal{P}}_n\} = -\frac{im\lambda}{2(R-\lambda Q_K)}\hat{\bar{\mathcal{Q}}}_m \delta_{n,0} \tag{2.50}$$

Their Poisson bracket with $E_R$ is similar to that of $\bar{\mathcal{P}}, \bar{\mathcal{Q}}$, (2.27). Their commutators with the modes of the currents are almost the usual ones, except the one with the zero mode of $\mathcal{P}$. However, if we replace $\mathcal{P}$ by $\tilde{\mathcal{P}}$, which amounts to modifying the zero mode in question, then they are exactly the usual ones. The spectral-flow-invariant generators satisfy certain homogenous flow equations, more precisely $\mathcal{D}_\lambda \hat{Q}_m = \widetilde{\mathcal{D}}_\lambda \hat{Q}_m = 0$ for the left-moving ones and $\widetilde{\mathcal{D}}_\lambda \hat{\bar{\mathcal{Q}}} = 0$ for the right-moving ones.

As for the flowed generators (2.26), since they satisfy a Witt-Kac-Moody algebra, they can automatically be written in the form (2.45) with the same $\hat{Q}_m, \hat{\bar{\mathcal{Q}}}_m$, but with $P_m, \bar{\mathcal{P}}_m$ replaced by $\widetilde{P}_m, \widetilde{\bar{\mathcal{P}}}_m$.

To summarize, all the left and right-moving generators, both flowed and unflowed, can be written as the sum of a spectral-flow invariant contribution and a Sugawara one. These expressions are classical. Of course, in the quantum theory, one needs to be careful about operator ordering issues. This is the subject of the next section.

# 3. The quantum symmetry algebra

## 3.1. Definitions of the quantum symmetry generators

A natural question is whether the classical symmetry generators described in the previous section can be promoted to quantum ones and if so, what is the symmetry algebra of the corresponding quantum generators. This can include, for example, central extensions associated to anomalies, as well as quantum corrections to the structure constants of the algebra.

The deformed generators will be denoted as $L_n^\lambda, K_n^\lambda, \bar{L}_n^\lambda$ and $\bar{K}_n^\lambda$. They are formally defined as[9]

$$L_n^\lambda \equiv R \, {}^\circ_\circ \int_0^R d\sigma \, e^{\frac{in(\sigma+t)}{R}} \mathcal{H}_L \, {}^\circ_\circ \, , \qquad K_n^\lambda \equiv {}^\circ_\circ \int_0^R d\sigma \, e^{\frac{in(\sigma+t)}{R}} (\mathcal{J}_+ + \frac{\lambda k}{2}\mathcal{H}_R) \, {}^\circ_\circ$$

$$\bar{L}_n^\lambda \equiv R_v \, {}^\circ_\circ \int_0^R d\sigma \, e^{-\frac{inv_{imp}}{R_v}} \mathcal{H}_R \, {}^\circ_\circ \, , \qquad \bar{K}_n^\lambda \equiv {}^\circ_\circ \int_0^R d\sigma \, e^{-\frac{inv_{imp}}{R_v}} (\mathcal{J}_- + \frac{\lambda k}{2}\mathcal{H}_R) \, {}^\circ_\circ \tag{3.2}$$

---

[9]The correspondence between the classical generators and the quantum ones is

$$RQ_m \leftrightarrow L_m^\lambda , \quad P_m \leftrightarrow K_m^\lambda , \quad R_v \bar{\mathcal{Q}}_m \leftrightarrow \bar{L}_m^\lambda , \quad \bar{\mathcal{P}}_m \leftrightarrow \bar{K}_m^\lambda , \quad E_R \rightarrow H_R \tag{3.1}$$

as well as their tilded and hatted counterparts. The map between Poisson brackets and commutators is $\{\,,\,\}_{P.B.} = -i[\,,\,]$.

i.e. they are given by the classical expressions (2.7), (2.21), subject to an appropriate ordering prescription and to quantum corrections, altogether denoted by '$\substack{\circ\\\circ}$'. This prescription can be established order by order in $\lambda$, after expanding the known fully non-linear expression (2.2) for the deformed classical symmetry currents in terms of the undeformed currents $\mathcal{H}_{L,R}^{(0)}, \mathcal{J}_\pm$. The first few terms in this expansion are

$$\mathcal{H}_{L,R}^\lambda(\sigma) = \mathcal{H}_{L,R}^{(0)}(\sigma) + \lambda \mathcal{J}_+(\sigma)\mathcal{H}_R^{(0)}(\sigma) + \lambda^2 \mathcal{H}_R^{(0)}(\sigma)\left(\mathcal{J}_+^2(\sigma) + \frac{k\mathcal{H}_R^{(0)}(\sigma)}{4}\right) + \dots \tag{3.3}$$

The prescription $\substack{\circ\\\circ}$ should be such that the resulting quantum generators are well defined and obey a sensible algebra. Note we have chosen to include an overall factor of $R_v$ in the definition of the right-moving pseudoconformal charges, in order to simplify their algebra. This implies, in particular, that the relation between the zero mode of the $\bar{L}_n^\lambda$ algebra and the right-moving Hamiltonian, $H_R$, is $\bar{L}_0^\lambda = R_v H_R$. The unrescaled generators are simply obtained by dividing by $R_v$, since $R_v = R - \lambda(J_0 - \bar{J}_0)$ commutes with all the charges.

Our main task is thus to determine the ordering prescription '$\substack{\circ\\\circ}$'. A first and rather non-trivial requirement is that the algebra of the resulting generators close. This requirement is insufficient, however, to fully fix the quantum generators and their commutation relations, as finite correction terms may be independently added to them to achieve e.g., a more familiar-looking algebra. The resulting symmetry algebra may thus appear arbitrary, and potentially correlated corrections in the different generators would be missed. Another drawback of this approach is that it is not systematic, in the sense that it is not clear *a priori* which operator ordering will generate a sensible, closed algebra.

**Flow equations**

A potentially better way to proceed is to use the quantum counterpart of the classical flow equations (2.31) that the symmetry generators satisfy to fix the quantum corrections. That the quantum counterpart exists follows from the very definition [25] of the operator $\tilde{\mathcal{O}}$ as the operator appearing in the flow that relates energy eigenstates in the various deformed theories[10]

$$\partial_\lambda |n_\lambda\rangle = \tilde{\mathcal{O}} |n_\lambda\rangle \tag{3.4}$$

The explicit expression for $\tilde{\mathcal{O}}$ follows from first order quantum-mechanical perturbation theory [25, 27] and is directly derived from the $J\bar{T}$ deforming operator $\mathcal{O}_{J\bar{T}}$, via [28]

$$[H, \tilde{\mathcal{O}}] = \mathcal{O}_{J\bar{T}}^{off-diag.} = \int d\sigma \, \epsilon^{\alpha\beta} J_\alpha T_{\beta V} + \sum_n \partial_\lambda E^{(n)} |n_\lambda\rangle\langle n_\lambda| \tag{3.5}$$

and the requirement that its diagonal elements in the energy eigenbasis vanish. The $J\bar{T}$ operator itself is defined via point splitting [15]. In the classical limit, $\tilde{\mathcal{O}}_{cls}$ is given by the expressions (2.29), (2.33), which hold on the $t = 0$ slice. If, moreover, one believes that $J\bar{T}$ - deformed CFTs on a cylinder exist[11] and the deformation does not affect the Hilbert space of states on the cylinder, but only the Hamiltonian and its associated eigenstates - which are smoothly modified as a function of $\lambda$ - one concludes that $\tilde{\mathcal{O}}$, given above, should be well defined at the full quantum level.

Given the full quantum operator $\tilde{\mathcal{O}}$, we can simply *define*

$$\partial_\lambda \widetilde{L}_m^\lambda \equiv [\tilde{\mathcal{O}}, \widetilde{L}_m^\lambda], \qquad \partial_\lambda \widetilde{K}_m^\lambda \equiv [\tilde{\mathcal{O}}, \widetilde{K}_m^\lambda]$$

$$\partial_\lambda \widetilde{\bar{L}}_m^\lambda \equiv [\tilde{\mathcal{O}}, \widetilde{\bar{L}}_m^\lambda], \qquad \partial_\lambda \widetilde{\bar{K}}_m^\lambda \equiv [\tilde{\mathcal{O}}, \widetilde{\bar{K}}_m^\lambda] \tag{3.6}$$

---

[10]Note that with this definition, $\tilde{\mathcal{O}}$ is anti-Hermitean.

[11]One potential worry is the fact that the energies of $J\bar{T}$ - deformed CFTs on a cylinder become imaginary at large enough initial dimension, for any fixed value of $\lambda/R$, suggesting that the finite-size theory is non-perturbatively ill-defined. This is due to the presence of superluminal modes, which in compact space give rise to closed timelike curves [29]. On the other hand, finite size provides a nice and simple set of observables to study that certainly make sense perturbatively in $\lambda$, to any desired order. It is in this restricted sense that we mean the well-definiteness of $J\bar{T}$ - deformed CFTs on a cylinder.

Since the flow equations are homogenous, it immediately follows (assuming that $\tilde{\mathcal{O}}$ satisfies all necessary Jacobi identities) that the $\widetilde{L}^\lambda_m$ etc. generators so defined satisfy the same algebra as at $\lambda = 0$, i.e. two commuting copies of the Virasoro-Kac-Moody algebra, with the same central extension, $c$, as that of the undeformed CFT. It is similarly easy to show [30] that energy eigenstates are also eigenstates of $\widetilde{L}^\lambda_0, \widetilde{\bar{L}}^\lambda_0$, with the same eigenvalue as in the undeformed CFT[12]. This construction is essentially the same as that of the spectrum-generating operators of [30], though it is not clear whether the authors considered their $\widetilde{L}^\lambda_m$ analogues to be *symmetry* generators, as we will argue at the end of this section.

Given that we were originally interested in the unflowed generators (3.2), we note that it may also be possible to use a flow equation to define the left-moving generators as

$$\partial_\lambda L^\lambda_m = [\mathcal{O}, L^\lambda_m], \qquad \partial_\lambda K^\lambda_m = [\mathcal{O}, K^\lambda_m] \tag{3.7}$$

where $\mathcal{O}$ equals (2.29) in the classical limit. At the quantum level, it may be possible to define $\mathcal{O}$ as the sum of an appropriate generalization of the two terms in (2.29), once all quantum corrections to the currents from which they are built are understood. Note that $\mathcal{O}$ does not, strictly speaking, correspond to a well-defined operator, due to the terms proportional to the zero modes $\chi_0, \phi_0$ it contains, which make its action on the Hilbert space ill-defined sometimes. However, its action on the left-moving generators does appear to make sense, which motivates the definition above. This definition immediately implies that $L^\lambda_m, K^\lambda_m$ obey a Virasoro-Kac-Moody algebra with central charge $c$, thus fulfilling our expectation.

Other generators that obey simple flow equations are the spectral-flow-invariant operators $\hat{L}^\lambda_m$ and $\hat{\bar{L}}^\lambda_m$, defined via

$$\hat{L}^\lambda_m = \widetilde{L}^\lambda_m - \frac{1}{k} \sum_\ell : \widetilde{K}^\lambda_{-\ell} \widetilde{K}^\lambda_{m+\ell} :, \qquad \hat{\bar{L}}^\lambda_m = \widetilde{\bar{L}}^\lambda_m - \frac{1}{k} \sum_\ell : \widetilde{\bar{K}}^\lambda_{-\ell} \widetilde{\bar{K}}^\lambda_{m+\ell} : \tag{3.8}$$

with $\widetilde{L}^\lambda_m$ etc., obtained by solving (3.6). It immediately follows that they satisfy

$$\tilde{\mathcal{D}}_\lambda \hat{L}^\lambda_m = \tilde{\mathcal{D}}_\lambda \hat{\bar{L}}^\lambda_m = 0 \tag{3.9}$$

where $\tilde{\mathcal{D}}_\lambda \equiv \partial_\lambda - [\tilde{\mathcal{O}}, \cdot]$. Conversely, one could start with $\hat{L}^\lambda_m, \widetilde{\mathcal{P}}^\lambda_m$ that satisfy the $\tilde{\mathcal{D}}_\lambda$ flow equation and show that the $\widetilde{L}^\lambda_m$ defined through the Sugawara procedure also satisfies the flow equation. As discussed in the sequel, the left-moving $\hat{L}^\lambda_m$ can be alternatively defined as solutions to $\mathcal{D}_\lambda \hat{L}^\lambda_m = 0$. Note that the "unflowed" right-moving generators $\bar{L}^\lambda_m$ are not as naturally defined via flow equations, because the latter have an inhomogenous piece that suffers from ordering ambiguities.

Of course, from a practical point of view, the exact quantum-mechanical expressions for $\tilde{\mathcal{O}}$ and $\mathcal{O}$ need to be calculated, since the classical expressions (2.29), (2.33) only determine them up to ordering ambiguities, as well as possible quantum corrections. The ordering of these operators is in principle fixed by their definition in terms of the $J\bar{T}$ deforming operator, which is well defined [15]; however, it can itself receive quantum corrections[13], which need to be incorporated. In addition, there could still be ambiguities in solving (3.5), especially when working in perturbation theory. Note, however, that these ambiguities are significantly fewer than individual ambiguities in each of the generators, and they lead to correlated corrections, which simultaneously affect all symmetry generators, without affecting the symmetry algebra - at least as long as $\tilde{\mathcal{O}}$ is a well-defined operator. These features will be exemplified in the following section, where we use the flow equations to explicitly compute the symmetry generators up to second order in the $J\bar{T}$ deformation parameter.

---

[12]Note this does not imply that $\widetilde{L}^\lambda_0, \widetilde{\bar{L}}^\lambda_0$ coincide with the undeformed left-/right-moving Hamiltonians $L_0, \bar{L}_0$ as operators: even if their eigenvalues are the same, their eigenvectors are clearly different.

[13]These quantum corrections can in principle be determined systematically, order by order in $\lambda$, by using the flow equations. That is, knowledge of the quantum-corrected $J\bar{T}$ operator up to a certain order would determine $\tilde{\mathcal{O}}$ up to the same order, using (3.5), which in turn can be used to determine the right-moving generators at the *next* order, using (3.6). These should in turn determine the next-order correction to the right-moving current $\mathcal{H}_R$, which enters the definition of the $J\bar{T}$ operator. The last step does not appear straightforward, however, because the right-moving generators are not simply the Fourier modes of $\mathcal{H}_R$, as in a usual CFT (3.32), and thus in $J\bar{T}$ - deformed CFTs, extracting $\mathcal{H}_R$ from the right-moving conserved charges is complicated, even though in principle, they encode the same information.

## Unflowed generators and the Sugawara construction

Given the flowed operators $\widetilde{L}_m^\lambda$, etc., one can construct the quantum counterparts of the standard "un-flowed" ones by appropriately generalising the classical relations (2.26) to the quantum case. For the affine $U(1)$ generators and the left-moving Virasoro ones, the natural, unambiguous generalisation is

$$K_m^\lambda \equiv \widetilde{K}_m^\lambda + \frac{\lambda k}{2} H_R \, \delta_{m,0} \,, \qquad \bar{K}_m^\lambda \equiv \widetilde{\bar{K}}_m^\lambda + \frac{\lambda k}{2} H_R \, \delta_{m,0} \tag{3.10}$$

$$L_m^\lambda \equiv \widetilde{L}_m^\lambda + \lambda \widetilde{K}_m H_R + \frac{k\lambda^2}{4} H_R^2 \, \delta_{m,0} \tag{3.11}$$

In the case of the left-movers, one should be able to check, in principle, whether the generators so defined coincide at the full quantum level with those obtained via the flow equation (3.7).

The situation is a bit more complicated in the case of the right-moving pseudoconformal generators, for which an ordering prescription is necessary. One may attempt to motivate one by using the Sugawara construction. For this, one can first construct the spectral-flow-invariant operators $\hat{L}_m^\lambda, \hat{\bar{L}}_m^\lambda$ at the full quantum level using the flow equation (3.9); note that for completeness, we have also included the left-moving generators in the discussion. The resulting generators would thus obey a Virasoro algebra with central charge $c - 1$, by construction. The unflowed (pseudo)conformal generators are then defined as

$$L_m^{\lambda,S} \equiv \hat{L}_m^\lambda + \frac{1}{k} \sum_\ell \, : K_{-\ell}^\lambda K_{m+\ell}^\lambda : \,, \qquad \bar{L}_m^{\lambda,S} \equiv \hat{\bar{L}}_m^\lambda + \frac{1}{k} \sum_\ell \, : \bar{K}_{-\ell}^\lambda \bar{K}_{m+\ell}^\lambda : \tag{3.12}$$

with ':' some normal ordering prescription and $K_m^\lambda, \bar{K}_m^\lambda$ given by (3.10), by assumption. Rewriting the above in terms of $\widetilde{L}_m^\lambda$ and $\widetilde{K}_m^\lambda$, we find that $L_m^{\lambda,S}$ coincides with $L_m^\lambda$ given in (3.11). As for the right-movers, if we use the normal ordering spelled out in (2.37), we find

$$\bar{L}_m^{\lambda,S} = \widetilde{\bar{L}}_m^\lambda + \begin{cases} \frac{\lambda}{2}(H_R \widetilde{\bar{K}}_m^\lambda + \widetilde{\bar{K}}_m^\lambda H_R) & if \quad m \geq 0 \\[2mm] \lambda \widetilde{\bar{K}}_m^\lambda H_R & if \quad m < 0 \end{cases} + \frac{k\lambda^2}{4} H_R^2 \, \delta_{m,0} \tag{3.13}$$

More generally, we will have

$$\bar{L}_m^\lambda = \widetilde{\bar{L}}_m^\lambda + \lambda \widetilde{\bar{K}}_m^\lambda H_R + \lambda \xi_m [H_R, \widetilde{\bar{K}}_m^\lambda] + \frac{k\lambda^2}{4} H_R^2 \delta_{m,0} \tag{3.14}$$

where $\xi_m$ depends rather sensitively on the ordering prescription used in (3.12). For example, another possible choice of normal ordering is

$$\sum_\ell \, : A_{-\ell} A_{m+\ell} :' \equiv \sum_{m+\ell > -\ell} A_{-\ell} A_{m+\ell} + \sum_{m+\ell \leq -\ell} A_{m+\ell} A_{-\ell} \tag{3.15}$$

It is easy to check that, using $:'$ instead of $:$ in e.g. (3.8) has no bearing on the tilded generator algebra, but only yields a constant shift in $\widetilde{\bar{L}}_0^\lambda$. On the other hand, if $A_m = \bar{K}_m^\lambda$, we obtain

$$\bar{L}_m^{\lambda,S'} = \widetilde{\bar{L}}_m^\lambda + \begin{cases} \lambda H_R \widetilde{\bar{K}}_m^\lambda & if \quad m \geq 0 \\[2mm] \lambda \widetilde{\bar{K}}_m^\lambda H_R & if \quad m < 0 \end{cases} + \frac{k\lambda^2}{4} H_R^2 \delta_{m,0} \tag{3.16}$$

i.e. $\xi_m = \Theta(m)$, which is different from (3.13). Thus, because $\widetilde{\bar{K}}_m^\lambda$ and $H_R$ do not commute, different normal ordering prescriptions will yield different expressions for the $\bar{L}_m^\lambda$, which will in turn result in different algebras for the unflowed generators, as we will show in more detail in section 3.3.

The flow equation obeyed by the generators (3.14) is

$$\tilde{\mathcal{D}}_\lambda \bar{L}_m^\lambda = \bar{K}_m^\lambda \tilde{\mathcal{D}}_\lambda (\lambda H_R) + \xi_m [\tilde{\mathcal{D}}_\lambda (\lambda H_R), \bar{K}_m^\lambda] \tag{3.17}$$

where $\tilde{\mathcal{D}}_\lambda H_R$ is computed below in (3.20). Thus, the ambiguity in the ordering of (3.12) translates into an ambiguity in the ordering of the inhomogenous term in the flow equation.

To fix this ambiguity, one may recall that the original CFT generators satisfy a Hermiticity condition, $\bar{L}_m^\dagger = \bar{L}_{-m}$. Using the fact that $\tilde{\mathcal{O}}$ is anti-Hermitean, one can show that this condition is preserved along the flow, namely that

$$(\bar{L}_m^\lambda)^\dagger = \bar{L}_{-m}^\lambda \tag{3.18}$$

Imposing this condition on (3.14) and using the fact that the $\widetilde{\bar{L}}_m^\lambda$ already satisfy this relation, as it follows from their definition and the anti-hermiticity of $\tilde{\mathcal{O}}$, one finds that the value of $\xi_m$ for which this condition is satisfied is the one given by the normal ordering (3.15), i.e. $\xi_m = \Theta(m)$.

Please note that even though (3.11), (3.14) are well-defined quantum-mechanical definitions of the unflowed symmetry generators, we do not currently have a way of checking whether this is what we mean by (3.2). It is envisageable that the full quantum definition of these operators could contain corrections to (3.11), (3.14) of higher order in $\hbar$, which would be invisible in the classical expressions. For the left-movers, we can check whether such corrections are present by using their alternate definition in terms of the flow equation (3.7), which contains in principle all quantum corrections to $\mathcal{O}$, and thus to the $L_m^\lambda$. This, however, does not so straightforwardly apply to the right-movers, because the inhomogenous terms in their flow equation could themselves receive quantum corrections at higher order in $\hbar$. Thus, unlike the $\widetilde{L}_m^\lambda$ and their barred counterparts, which are in principle well-defined at the full quantum level, the unflowed generators appear, at least from the perspective of this particular construction, as a less natural basis of symmetry generators.

What makes the unflowed generators special is, of course, the fact that their zero mode is simply related to the left- and, respectively, right-moving Hamiltonian. Concretely[14], we have

$$L_0^\lambda = H_L = \widetilde{L}_0^\lambda + \lambda J_0 H_R + \frac{k\lambda^2}{4}H_R^2 \,, \qquad \bar{L}_0^\lambda = R_v H_R = \widetilde{\bar{L}}_0^\lambda + \lambda \bar{J}_0 H_R + \frac{k\lambda^2}{4}H_R^2 \tag{3.19}$$

According to an argument given in [30], this relation is expected to be exact, because it follows from the universal expression for the finite-size spectrum of of $J\bar{T}$ - deformed CFTs on a cylinder, which is believed to be an exact formula. Note this implies in particular that

$$\widetilde{\mathcal{D}}_\lambda H_R = \frac{H_R Q_K}{R - \lambda Q_K} \tag{3.20}$$

at the full quantum level, where we used (3.6). Whether the remaining $\bar{L}_m^\lambda$ receive further quantum corrections beyond (3.14) is still an open question, as we already discussed.

**Conservation**

The final question that we would like to address in this section is why the flowed generators $\widetilde{L}_m^\lambda$ etc., which are defined to satisfy the same flow equation as the energy eigenstates, should correspond to *symmetries* of the system. The reader might be rightfully skeptical of this definition, which appears devoid of any physical content: indeed, any (integrable) deformation of the theory, including massive ones, that smoothly deforms the states of the system will admit a Virasoro algebra, according to this formal construction.

In order for the Virasoro generators to actually correspond to symmetries of the system, one should additionally be able to construct an infinite number of *conserved quantities* from them. In the classical theory, this is indeed the case, and we provided the explicit form (2.7), (2.21) of the conserved charges. We would now like to argue that also in the quantum theory, it is possible to add an explicit time dependence to the generators (which matches the time dependence of the classical generators in the $\hbar \to 0$ limit), such that they satisfy the conservation requirement (1.3).

---

[14]The relationship between $H_R$ and $\widetilde{\bar{L}}_0^\lambda$ appears very similar to that between $\mathcal{H}_R$ and $\mathcal{H}_R^{(0)}$, given in (2.2). One should note though that (2.2) is a relation between local currents, whereas (3.19) is a relation between their zero modes, which does not follow in any simple way from the local relation. See also comments in the previous footnote.

For this, we need to know the commutation relations of the flowed generators with the Hamiltonian, $H = H_L + H_R = 2H_R + P$. The commutators of the left-moving generators with the Hamiltonian are

$$[H, \widetilde{L}_m^\lambda] = [H_L + H_R, \widetilde{L}_m^\lambda] = -\frac{m\hbar}{R}\widetilde{L}_m^\lambda \tag{3.21}$$

where we used the expression (3.19) for $H_L = L_0^\lambda$. Even though this does not vanish, it implies that conjugating $L_m^\lambda$ by the time evolution operator will simply yield $L_m^\lambda$ multiplied by a c-number function of time. By adding an explicit time dependence to the Schrödinger-picture generator, of the form

$$\widetilde{L}_{m,S}^\lambda(t) = e^{\frac{imt}{R}}\widetilde{L}_{m,S}^\lambda(0) \tag{3.22}$$

one obtains an operator whose expectation value is conserved, i.e., the Heisenberg picture operator satisfies (1.3). Same holds for $\widetilde{K}_m^\lambda$.

The commutation relations of the $\widetilde{\widetilde{L}}_m^\lambda, \widetilde{\widetilde{K}}_m^\lambda$ with the Hamiltonian can be computed by assuming they take the following form

$$[\widetilde{\widetilde{L}}_m^\lambda, H_R] = \widetilde{\widetilde{L}}_m^\lambda \alpha_m^r = \alpha_m^l \widetilde{\widetilde{L}}_m^\lambda \tag{3.23}$$

inspired by the classical Poisson brackets (2.27), where $\alpha_m^{l,r}$ are functions of the right-moving Hamiltonian and the global $U(1)$ charges. Note that $\alpha_m^{l,r}$ need not commute with $\widetilde{\widetilde{L}}_m^\lambda$, but do commute with $H_R$, and can in principle be different. Their explicit expressions, worked out in appendix A, follow from the commutation relations of the $\widetilde{\widetilde{L}}_m^\lambda$ and the expression (3.19) for its zero mode in terms of $H_R$. They read

$$\alpha_m^{l,r}(H_R) = \pm 2\frac{R - \lambda Q_K - \sqrt{(R - \lambda Q_K)^2 \mp \hbar km\lambda^2}}{k\lambda^2} \approx \frac{m\hbar}{R - \lambda Q_K} \pm \frac{km^2\lambda^2\hbar^2}{4(R - \lambda Q_K)^3} + \mathcal{O}(\hbar^3) \tag{3.24}$$

where $Q_K = J_0 + \frac{\lambda k}{2}H_R$. This reproduces the classical result (2.27) to leading order in $\hbar$. One can alternatively write $\alpha_m^{l,r}$ in terms of $\widetilde{\widetilde{L}}_0^\lambda$, whose eigenvalues are simply the right-moving dimensions in the undeformed CFT, by using the relation

$$R - \lambda Q_K = \sqrt{(R - \lambda J_0)^2 - \lambda^2 k\widetilde{\widetilde{L}}_0^\lambda} \tag{3.25}$$

If the eigenvalue of $\widetilde{\widetilde{L}}_0^\lambda$ is such that the square root above is real, then note that $\alpha_m^l$ is well-defined for large negative $m$, whereas $\alpha_m^r$ is well-defined for large $m > 0$. Otherwise, one can simply see (3.24) as perturbative expressions, which hold up to any desired order in $\lambda$.

The commutator of $\widetilde{\widetilde{K}}_m^\lambda$ with $H_R$ is similarly given by

$$[\widetilde{\widetilde{K}}_m^\lambda, H_R] = \widetilde{\widetilde{K}}_m^\lambda \alpha_m^r = \alpha_m^l \widetilde{\widetilde{K}}_m^\lambda \tag{3.26}$$

Using certain identities derived in the appendix, one can show that the relation between $\alpha_m^l$ and $\alpha_m^r$ is consistent with the relation (3.23) between them, and that all the Jacobi identities involving $\widetilde{\widetilde{L}}_m^\lambda, \widetilde{\widetilde{K}}_m^\lambda$ and $H_R$ are satisfied.

The commutator of the right-moving generators with the full Hamiltonian $H = 2H_R + P$, where $P = H_L - H_R = (\widetilde{L}_0^\lambda - \widetilde{\widetilde{L}}_0^\lambda)/R$ is

$$[H, \widetilde{\widetilde{L}}_m^\lambda] = \left(\frac{m\hbar}{R} - 2\alpha_m^l\right)\widetilde{\widetilde{L}}_m^\lambda \tag{3.27}$$

In analogy to the procedure employed in the case of the usual conformal transformations, one can simply include an explicit time dependence in the Schrödinger-picture generator

$$\widetilde{\widetilde{L}}_{m,S}^\lambda(t) = \widetilde{\widetilde{L}}_{m,S}^\lambda(0)\, e^{i(\frac{2\alpha_m^r}{\hbar} - \frac{m}{R})t} = e^{i(\frac{2\alpha_m^l}{\hbar} - \frac{m}{R})t}\widetilde{\widetilde{L}}_{m,S}^\lambda(0) \tag{3.28}$$

to make its expectation value conserved at the full quantum level. We thus conclue that the infinite number of generators $\widetilde{\bar{L}}_m^\lambda$ defined via the flow equations (3.6) correspond to *symmetries* of the system, since we have just explicitly constructed an infinite set of conserved quantities from them.

One can check, using the special properties of the functions $\alpha_m^{l,r}$, that the algebra of the $\widetilde{\bar{L}}_m^\lambda$ is time-independent, as expected. The time dependence that one needs to attach to the right-moving flowed current generators $\widetilde{\bar{K}}_m^\lambda$, as well as to the unflowed generators $\bar{L}_m^\lambda, \bar{K}_m^\lambda$ in order for them to satisfy (1.3) is identical to the one above. It is interesting to compare this time dependence to that of the classical expressions (3.2), as one can easily note the presence of an infinite series of corrections in $\hbar$. It is interesting to ask whether (3.2) receives any additional quantum corrections.

**Summary**

The upshot of this discussion is that the definition of the $\widetilde{\bar{L}}_m^\lambda$, etc. via the flow equation, together with the fact that their zero modes (3.19) are simple, known functions of the Hamiltonian of the system, allows one to show that $\widetilde{\bar{L}}_m^\lambda$ etc., correspond to an infinite set of conserved charges, which implies they are *symmetries* of $J\bar{T}$ - deformed CFTs. These symmetries organise into two commuting copies of a Virasoro-Kac-Moody algebra. This argument is extremely general, and it immediately applies to other integrable deformations, such as the $T\bar{T}$ one; see section (4) for details.

If the discussion in this section may have appeared somewhat abstract, in the following one we embark on a very explicit calculation of the corrections to the quantum generators up to $\mathcal{O}(\lambda^2)$, using the various definitions we provided. Readers not interested in the technical details can skip directly to section 3.3.

## 3.2. Perturbative construction of the quantum symmetry generators

To proceed, let us first set up the notation. The undeformed symmetry generators will be denoted as $L_m, J_m, \bar{L}_m, \bar{J}_m$, defined via

$$L_n = R \int_0^R e^{\frac{in\sigma}{R}} \mathcal{H}_L^{(0)}(\sigma) \,, \qquad J_n = \int_0^R e^{\frac{in\sigma}{R}} \mathcal{J}_+(\sigma) \tag{3.29}$$

$$\bar{L}_n = R \int_0^R d\sigma e^{-\frac{in\sigma}{R}} \mathcal{H}_R^{(0)}(\sigma) \,, \qquad \bar{J}_n = \int_0^R d\sigma e^{-\frac{in\sigma}{R}} \mathcal{J}_-(\sigma) \tag{3.30}$$

where $\mathcal{H}_{L,R}^{(0)}, \mathcal{J}_\pm$ now represent operators in the undeformed CFT. We have included an overall factor of $R$, so that the generators be dimensionless. The (quantum) commutation relations of these modes are the standard ones, given in (2.35). The inverse relations read

$$\mathcal{H}_L^{(0)} = \sum_n \frac{L_n}{R^2} e^{-in\sigma/R} \,, \qquad \mathcal{J}_+ = \sum_n \frac{J_n}{R} e^{-in\sigma/R} \tag{3.31}$$

$$\mathcal{H}_R^{(0)} = \sum_n \frac{\bar{L}_n}{R^2} e^{in\sigma/R} \,, \qquad \mathcal{J}_- = \sum_n \frac{\bar{J}_n}{R} e^{in\sigma/R} \tag{3.32}$$

The generators above are defined on the $t = 0$ slice. If we decided to work at $t \neq 0$, we would simply replace $\sigma \to \sigma + t$ in the left-moving generators and $\sigma \to \sigma - t$ in the right-moving ones. In any case, these generators are time-independent.

We will also need the expansion of $\phi$ on the $t = 0$ slice, which is given by

$$\phi(\sigma) = \phi_0 + \frac{w\sigma}{R} + \hat{\phi}_{nzm} = \phi_0 + \frac{\sigma}{R}(J_0 - \bar{J}_0) + i \sum_{m \neq 0} \frac{1}{m}(J_m - \bar{J}_{-m}) e^{-\frac{im\sigma}{R}} \tag{3.33}$$

where we have separated out the winding from the zero mode and the non-zero ones. The commutation relations of the zero mode are

$$[L_n, \phi_0] = -iJ_n \,, \qquad [\bar{L}_n, \phi_0] = -i\bar{J}_n \,, \qquad [J_n, \phi_0] = -\frac{ik}{2}\delta_{n,0} \,, \qquad [\bar{J}_n, \phi_0] = -\frac{ik}{2}\delta_{n,0} \tag{3.34}$$

Note that the expansion of $\mathcal{J}_\pm(\sigma)$ and $\phi(\sigma)$ on the $t = 0$ slice will be the same in the deformed theory as in the undeformed one, by definition. However, in the deformed theory, time evolution to $t \neq 0$ will be implemented by a much more complicated operator, so the expression for these fields on a later slice will not be as easily obtained from the $t = 0$ ones as above. In this section we will therefore construct the expansion of the quantum generators on the $t = 0$ slice only, and simply rely on the general result (3.22), (3.28) for their expression at a different time. We also set $R = 1$ thoughout.

**The left-moving generators**

Our discussion of the left-moving generators will be rather detailed, in order to exemplify how the calculations are performed and highlight the technical issues that arise. The expression for the deformed left-moving generators is given in (3.2). The expansion of $\mathcal{H}_L$ in powers of $\lambda$ is given up to $\mathcal{O}(\lambda^2)$ in (3.3), while that of the holomorphic current is

$$\mathcal{J}_+ + \frac{\lambda k}{2}\mathcal{H}_R = \mathcal{J}_+ + \frac{\lambda k}{2}\mathcal{H}_R^{(0)} + \frac{\lambda^2 k}{2}\mathcal{J}_+\mathcal{H}_R^{(0)} + \dots \tag{3.35}$$

Plugging in the peturbative expansion of the currents into the generators, performing the Fourier transform and promoting the classical fields to quantum operators, we obtain the following "uncorrected" expressions

$$L_n^{\lambda,uncorr} = L_n + \lambda \sum_m \bar{L}_m J_{n+m} + \lambda^2 \sum_{m,\ell} \bar{L}_m J_\ell J_{n+m-\ell} + \frac{\lambda^2 k}{4} \sum_m \bar{L}_m \bar{L}_{-n-m} + \dots \tag{3.36}$$

$$K_m^{\lambda,uncorr} = J_m + \frac{\lambda k}{2}\left( \bar{L}_{-m} + \lambda \sum_\ell J_{m+\ell}\bar{L}_\ell + \dots \right) \tag{3.37}$$

for which we now need to provide a $\overset{\circ}{\circ}$ prescription. Note that there are no ordering ambiguities up to $\mathcal{O}(\lambda)$ in $L_n^\lambda$, and up to $\mathcal{O}(\lambda^2)$ in $K_n^\lambda$. Our current task is thus to determine an appropriate ordering of the $\mathcal{O}(\lambda^2)$ terms in (3.36), as well as potential quantum corrections to both formulae.

Let us start by analysing the $K_m^\lambda$. The algebra of the uncorrected operators is

$$[K_m^{\lambda,uncorr}, K_n^{\lambda,uncorr}] = \frac{mk}{2}\delta_{m+n} - \frac{\lambda^2 c\, k^2}{48}m(m^2-1)\delta_{m+n} \tag{3.38}$$

where $c$ is the central charge of the undeformed CFT. While the generators (3.37) are certainly well-defined up to this point, we do however expect the algebra of the $K_m^\lambda$ to be simply Kac-Moody, as part of the usual enhancement of the global left-moving $SL(2,\mathbb{R}) \times U(1)$ symmetry of $J\bar{T}$ - deformed CFTs. It is thus highly desirable to absorb the unexpected second term into a shift

$$K_m^{\lambda,uncorr} \rightarrow K_m^{\lambda,corr} = K_m^{\lambda,uncorr} + \frac{\lambda^2 c\, k}{48}(m^2-1)J_m \tag{3.39}$$

One could in principle also add $\mathcal{O}(\lambda^2)$ terms that commute with $J_m$, which cannot be fixed by just this commutator. Since this correction is proportional to the central charge, it could not have been predicted by the classical analysis.

Our choice to shift the current in order to obtain the desired algebra may however seem arbitrary, so below we would like to show that it is in fact implemented naturally by the solution to the flow equation (3.6) or (3.7) for the Kac-Moody generators. Since the flow operator $\tilde{\mathcal{O}}$ entering (3.6) is better defined, we will start by analysing this equation which, according to (3.10), will construct for us all components of $K_m^\lambda$, except $m = 0$.

The operator $\tilde{\mathcal{O}}$ is defined directly in terms of the $J\bar{T}$ deforming operator via (3.5). An important point is that the $J\bar{T}$ operator defined in [15] is *not* normal ordered, i.e. its perturbative expansion in terms of modes is

$$\mathcal{O}_{J\bar{T}} = \epsilon^{\alpha\beta} J_\alpha T_{\beta V} = -2\left( \sum_m J_m \bar{L}_m + 2\lambda \sum_{m,p} J_m J_p \bar{L}_{m+p} + \frac{\lambda k}{2}\sum_m \bar{L}_{-m}\bar{L}_m + \dots \right) \tag{3.40}$$

where the ... contain higher corrections in $\lambda$, as well as potential quantum corrections at $\mathcal{O}(\lambda)$. In deriving this expansion, we have simply placed the mode expansion of $J_\alpha$ to the left of the mode expansion of $T_{\beta V}$. The classical expressions for all needed components can be found in [25]. The result is that the last two terms, which could in principle have had an ordering that depended on the sign of the mode number, do not, which appears to be related to the special factorization properties of the $J\bar{T}$ operator. The fact that the perturbative expansion of the $J\bar{T}$ operator is *not* creation-annihilation ordered is important in showing, for example, that the expectation value of the $J\bar{T}$ operator in energy eigenstates is finite[15].

Since we are interested in corrections up to $\mathcal{O}(\lambda^2)$ in $K_m^\lambda$, we need the expansion of $\tilde{\mathcal{O}}$ up to $\mathcal{O}(\lambda)$. Remember that $\tilde{\mathcal{O}}$ consists of two parts, $\hat{\mathcal{O}}$ and $\Delta\tilde{\mathcal{O}}$, whose classical limits are (2.29) and, respectively, (2.33). Given these classical limits and the definition (3.5), a natural proposal for the quantum operators $\hat{\mathcal{O}}$ and $\Delta\tilde{\mathcal{O}}$ up to $\mathcal{O}(\lambda)$ is given by

$$\hat{\mathcal{O}} = \frac{1}{2}\left(\sum_{\ell \neq 0}\frac{1}{\ell}(J_\ell - \bar{J}_{-\ell})(\bar{L}_\ell + \lambda\sum_p J_p\bar{L}_{\ell+p} + \ldots) - \text{ h.c.}\right)$$

and

$$\Delta\tilde{\mathcal{O}} = \frac{1}{2}\left(w\lambda\sum_{\ell \neq 0}\frac{1}{\ell}(J_\ell - \bar{J}_{-\ell})\bar{L}_\ell + \lambda\sum_{\ell \neq 0}\frac{1}{\ell}(J_\ell - \bar{J}_{-\ell})\bar{J}_\ell\bar{L}_0 - \text{h.c.}\right) \tag{3.42}$$

where 'h.c' denotes hermitean conjugation. This last term is needed in order to ensure that the full $\tilde{\mathcal{O}}$ is anti-hermitean. The operators above reproduce the necessary quantum correction (3.41) to the $J\bar{T}$ deforming operator[16]. Despite not being normal ordered, the expectation value of $\tilde{\mathcal{O}}$ in energy eigenstates is finite - more precisely, zero - by construction.

It is interesting to note that the expression for $\tilde{\mathcal{O}}$ given above is not the only possible solution to (3.5) up to this order. In fact, if one uses a creation-annihilation normal-ordered $\tilde{\mathcal{O}}$, which equals (twice) the first term in (3.42) for $\ell > 0$, and (twice) the hermitean conjugate term for $\ell < 0$, one still satisfies (3.5) for the *same* $J\bar{T}$ operator (3.40). It is interesting to investigate what is the impact of this alternate choice of flow operator, which we will denote as $: \tilde{\mathcal{O}} :$, on the symmetry generators and their algebra[17]. We will thus keep both options for now, and derive their respective consequences. Of course, other orderings may also be compatible with (3.5) at this order in perturbation theory, but we will not attempt to be exhaustive. One should keep in mind, however, that the non-normal-ordered option is a priori preferred.

The current $K_m^\lambda$ can be built iteratively, stating from $J_m$ at $\lambda = 0$. At zeroth order, $\tilde{\mathcal{O}} = \hat{\mathcal{O}}$, and the solution to the flow equation (3.6) is

$$\widetilde{K}_m^\lambda = K_m^\lambda = J_m + \frac{\lambda k}{2}\bar{L}_{-m} , \quad m \neq 0 \tag{3.43}$$

---

[15]To see this, one can take for example a deformed primary state at $\mathcal{O}(\lambda)$, which is given by

$$|h\rangle_\lambda = |h\rangle - \lambda\sum_{\ell=1}^{\infty}\frac{1}{\ell}J_{-\ell}\bar{L}_{-\ell}|h\rangle + \ldots$$

where we assumed $\tilde{\mathcal{O}}$, too, is normal ordered. If the terms in (3.40) were creation-annihilation ordered, then the expectation value $_\lambda\langle h|\mathcal{O}_{J\bar{T}}|h\rangle_\lambda$ would be infinite at $\mathcal{O}(\lambda)$. On the other hand, one can easily check that the non-normal ordered expression (3.40) does produce a finite answer, up to a term proportional to $c$. More precisely

$$_\lambda\langle h|\mathcal{O}_{J\bar{T}}|h\rangle_\lambda = \frac{\lambda kc}{12}\sum_{\ell=1}^{\infty}\ell(\ell^2 - 1) \tag{3.41}$$

This term can be absorbed into an $\mathcal{O}(\lambda c)$ correction to the $J\bar{T}$ operator (3.40).

[16]It is a valid question whether $\tilde{\mathcal{O}}$ itself receives quantum corrections proportional to $c$ at order $\lambda$, as we have found was the case for the $J\bar{T}$ operator. In principle, this question can be answered systematically by computing the $\mathcal{O}(\lambda)$ correction to all the generators and plugging it back into the components of the stress tensor that enter the $J\bar{T}$ deformation. In practice, this is somewhat complicated to implement because, unlike in usual CFTs (3.32), $\mathcal{H}_R$ is not simply related to the conserved right-moving charges via a Fourier transform. Given that (3.42) already produces the needed correction (3.41) to the $J\bar{T}$ operator, we will assume that no further corrections are necessary at this order.

[17]The question motivating this exercise is: if we made a mistake in computing $\tilde{\mathcal{O}}$, (how) would we discover it at the level of the generators? This question is non-trivial because *a priori* any choice of $\tilde{\mathcal{O}}$ will lead to a Virasoro-Kac-Moody algebra by construction, and the only way this will fail is if $\tilde{\mathcal{O}}$ does not, for example, satisfy the Jacobi identity.

and $\widetilde{K}_0^\lambda = J_0$ for $m = 0$. It is easy to see that this solution for the zero mode will hold to all orders in $\lambda$, since neither $\hat{\mathcal{O}}$, nor $\Delta\tilde{\mathcal{O}}$ contain any terms that do not commute with $J_0$. The first order solution for $K_m^\lambda$ is also given by the above, but now for all $m$. Plugging it into the flow equation at the next order, we obtain to $\mathcal{O}(\lambda)$

$$[\hat{\mathcal{O}}, K_m^\lambda] = \left(\frac{k}{2}\bar{L}_{-m} + \frac{\lambda k}{2}\sum_\ell J_\ell\bar{L}_{\ell-m} + \frac{\lambda kc}{24}(m^2-1)(J_m - \bar{J}_{-m})\right)_{m\neq 0} + \frac{\lambda k}{2}\sum_{\ell\neq 0} J_\ell\bar{L}_{\ell-m} +$$
$$+ \frac{\lambda k}{4}(2\bar{J}_0\bar{L}_{-m} - \bar{J}_{-m}\bar{L}_0 - \bar{L}_0\bar{J}_{-m}) \tag{3.44}$$

and

$$[\Delta\tilde{\mathcal{O}}, K_m^\lambda] = \frac{\lambda k}{4}(2w\bar{L}_{-m} + \bar{J}_{-m}\bar{L}_0 + \bar{L}_0\bar{J}_{-m}) , \qquad m \neq 0 \tag{3.45}$$

Adding up the two contributions and integrating with respect to $\lambda$, we find

$$K_m^\lambda = J_m + \frac{\lambda k}{2}\bar{L}_{-m} + \frac{\lambda^2 k}{2}\sum_\ell J_\ell\bar{L}_{\ell-m} + \frac{\lambda^2 kc}{48}(m^2-1)(J_m - \bar{J}_{-m})\bigg|_{m\neq 0} \tag{3.46}$$

Strictly speaking, we have only shown this for $m \neq 0$, since only these components satisfy $\tilde{\mathcal{D}}_\lambda K_m^\lambda = 0$. However, comparison with the uncorrected expression (3.37) indicates that this expression will hold for all $m$. We will also rederive it in the sequel, using the flow equation (3.7).

Had we used $:\tilde{\mathcal{O}}:$ instead to flow, we would have obtained an additional shift

$$K_m^\lambda(:) = K_m^\lambda + \frac{\lambda^2 km(m-1)}{8}\bar{J}_{-m} \tag{3.47}$$

due to the normal ordering prescription. This term does not affect the $K_m^\lambda$ commutator, but does contribute to the one with $\bar{K}_n^\lambda$.

We can also compare the solution (3.46) to that of the flow equation generated by $\mathcal{O}$, which has the advantage that it involves $K_m^\lambda$ for all $m$. For this, we need the quantum expression for the operator $\Delta\mathcal{O}$

$$\Delta\mathcal{O} = \frac{i}{2}(w\chi_0^\lambda - \phi_0 H_R) - \text{h.c.} \tag{3.48}$$

This Ansatz is based on the classical expression (2.29) and the ordering of the $J\bar{T}$ operator. The superscript of $\chi_0^\lambda$ indicates this quantity receives corrections at higher orders in $\lambda$, as compared with $\phi_0$, which does not. The right-moving Hamiltonian entering (3.48) is, to this order

$$H_R = \bar{L}_0 + \lambda\sum_p J_p\bar{L}_p + \dots \tag{3.49}$$

The commutator $[\chi_0^\lambda, K_m^\lambda] = -i\lambda k/2\bar{L}_{-m} + \dots$ follows from the Poisson bracket computed in [25]. Plugging all this into (3.48), we find

$$[\Delta\mathcal{O}, K_m^\lambda] = \frac{\lambda k}{2}(J_0 - \bar{J}_0)\bar{L}_{-m} + \frac{k}{2}\left(\bar{L}_0 + \lambda\sum_\ell J_\ell\bar{L}_\ell\right)\delta_{m,0} + \frac{\lambda k}{4}(\bar{J}_{-m}\bar{L}_0 + \bar{L}_0\bar{J}_{-m}) \tag{3.50}$$

Adding this up to (3.44), we obtain exactly the same expression (3.46) up to $\mathcal{O}(\lambda^2)$, now for all $m$. This exercise also serves as a cross check that the only difference between $K_m^\lambda$ and $\widetilde{K}_m^\lambda$, as defined via their respective flow equations, is indeed consistent with (3.10).

The commutator of the corrected $K_m^\lambda$ is simply

$$[K_m^\lambda, K_n^\lambda] = \frac{mk}{2}\delta_{m+n} + \mathcal{O}(\lambda^3) \tag{3.51}$$

as expected from our general arguments in the previous section.

Let us now turn to the $L_m^\lambda$, given in (3.36). To first order in $\lambda$, there is no ordering ambiguity and it can be easily checked that the generators satisfy the Virasoro algebra with central extension $c$. Understanding the appropriate ordering is however necessary at $\mathcal{O}(\lambda^2)$. As discussed in section 3.1, there exist at least two ways of defining the left-moving conformal generators. The simplest way to construct a well-defined $L_m^\lambda$ while avoiding to explicitly deal with the ordering is to use the Sugawara construction (3.12). If the $\hat{L}_m^\lambda$ and the $K_m^\lambda$ are well-defined and satisfy the expected algebra, then the well-definiteness and the algebra of the $L_m^\lambda$ follows.

As we saw in the previous section, at the classical level the spectral-flow-invariant left generators in the deformed theory are identical to the undeformed ones (2.47). The most natural generalization of this observation to the quantum level is to take

$$\hat{L}_m^\lambda = \hat{L}_m , \qquad \forall \lambda \tag{3.52}$$

This in particular implies that the $\hat{L}_m^\lambda$ satisfy a Virasoro algebra with central charge $c-1$. The above equality also follows from solving the flow equation $\tilde{\mathcal{D}}_\lambda \hat{L}_m^\lambda = 0$, since $\hat{L}_m$ commutes with all the operators that appear in the definition of $\tilde{\mathcal{O}}$. For the same reason, $\hat{L}_m^\lambda$ entirely commute with $K_n^\lambda$,

$$[\hat{L}_m^\lambda, K_n^\lambda] = 0 , \qquad \forall \lambda \tag{3.53}$$

The deformed left-moving generators obtained via the Sugawara construction then satisfy a Virasoro algebra with central charge $c$, since the building blocks satisfy (2.38). They also satisfy the flow equation $\mathcal{D}_\lambda L_m^\lambda = 0$, since both $K_m^\lambda$ and $\hat{L}_m^\lambda$ do. The ordering we chose for the Sugawara construction is automatically giving us a well-defined ordering prescription for the generators, which take the explicit form

$$
\begin{aligned}
L_n^{\lambda,S} &= L_n + \lambda \sum_\ell J_{n+\ell}\bar{L}_\ell + \frac{\lambda^2 k}{4}\left(\sum_{\ell>0}\bar{L}_\ell\bar{L}_{-n-\ell} + \sum_{\ell\geq0}\bar{L}_{\ell-n}\bar{L}_{-\ell}\right) + \frac{\lambda^2}{2}\sum_{\ell>0,p}(J_{-\ell}J_{n+\ell+p} + J_{-\ell+p}J_{n+\ell})\bar{L}_p + \\
&+ \frac{\lambda^2}{2}\sum_{\ell\geq0,p}(J_{n-\ell}J_{\ell+p} + J_{n-\ell+p}J_\ell)\bar{L}_p + \frac{\lambda^2 c}{48}\left(\sum_{\ell>0}(\ell^2-1)(J_{-\ell}-\bar{J}_\ell)J_{n+\ell} + \sum_{\ell\geq0}(\ell^2-1)J_{n-\ell}(J_\ell-\bar{J}_{-\ell})\right) + \\
&+ \frac{\lambda^2 c}{48}\left(\sum_{\ell>0}((n+\ell)^2-1)J_{-\ell}(J_{n+\ell}-\bar{J}_{-n-\ell})_{\ell\neq-n} + \sum_{\ell\geq0}((n-\ell)^2-1)(J_{n-\ell}-\bar{J}_{\ell-n})_{\ell\neq n}J_\ell\right)
\end{aligned}
\tag{3.54}
$$

Note that this does *not* correspond to the usual creation-annihilation normal ordering. In particular, the terms in the first paranthesis are anti-ordered.

One can alternatively construct the $L_m^\lambda$ by integrating the flow equation (3.7). If one uses the non-normal-ordered $\mathcal{O}$ to flow the seed $L_m$, one finds

$$\partial_\lambda L_m^\lambda = \sum_\ell J_{\ell+m}\bar{L}_\ell + \frac{\lambda k}{2}\sum_\ell \bar{L}_{-\ell-m}\bar{L}_\ell + 2\lambda \sum_{\ell,q} J_\ell J_{q+m}\bar{L}_{\ell+q} + \frac{\lambda c}{24}\sum_{\ell\neq0}(\ell^2-1)[(J_\ell-\bar{J}_{-\ell})J_{m-\ell} + J_{m-\ell}(J_\ell-\bar{J}_{-\ell})] \tag{3.55}$$

Since $L_m^{\lambda,S}$ also obeys the flow equation, the two end results must be the same, by consistency. Indeed, (3.55) can be shown to be equivalent to the Sugawara precription (3.54), through subtle cancellations of infinite terms. One can similarly show that the $L_m^{\lambda,S}(:)$ constructed via the Sugawara procedure, using $K_m^\lambda(:)$ instead of $K_m^\lambda$, is equivalent to the $L_m^\lambda(:)$ generated via the flow equation driven by $:\mathcal{O}:$. Thus, despite the slight ordering ambiguity we have decided to keep at this stage, the algebra of the left-moving generators is always Virasoro-Kac-Moody, with the same central charge as that of the original CFT. This fact follows, of course, from the definition of the generators via a homogenous flow equation, which will produce a Virasoro-Kac-Moody algebra for any choice of operator $\mathcal{O}$ that satisfies the Jacobi identity.

Given that the unflowed left-moving generators $L_m^\lambda$ can be defined by flowing the seed $L_m$ using (3.7), while the flowed generators $\widetilde{L}_m^\lambda$ are defined by flowing the same seed with $\tilde{\mathcal{O}}$, we can directly check the proposed relation (3.11) between the two up to $\mathcal{O}(\lambda^2)$, but up to any order in $\hbar$. At the first order, we obtain

$$\widetilde{L}_m^\lambda = L_m^\lambda - \lambda J_m \bar{L}_0 \tag{3.56}$$

while at the next order, we find

$$\partial_\lambda \widetilde{L}_m^\lambda = \partial_\lambda L_m^\lambda - J_m(\bar{L}_0 + 2\lambda \sum_p J_p \bar{L}_p) - \lambda k \bar{L}_{-m} \bar{L}_0 + \frac{\lambda k}{2} \bar{L}_0^2 \delta_{m,0} \tag{3.57}$$

which is precisely consistent with (3.11).

Finally, it is instructive to contrast the algebra of the generators obtained via the Sugawara construction or by integrating the flow equation with the algebra of a set of generators defined using a simple-minded creation-annihilation normal ordering for the operators $L_m^{\lambda,uncorr}$. The commutator of two : : ordered left generators is[18]

$$[: L_m^{\lambda,uncorr} :, : L_n^{\lambda,uncorr} :] = (m-n)\left[: L_{m+n}^\lambda : -\frac{k\lambda^2}{4}(m+n)\bar{L}_{-m-n}\sum_1^\infty 1 + \frac{\lambda^2 k}{2}\bar{L}_{-m-n}(m^2+n^2-mn-1)\right] +$$

$$+ \frac{\lambda^2 c}{12}\sum_\ell \ell(\ell^2-1)J_{m+\ell}J_{n-\ell} + \frac{c}{12}\left(m(m^2-1) - \sum_1^\infty \ell^2(\ell^2-1) + m\sum_1^\infty \ell(\ell^2-1)\right)\delta_{m+n} \tag{3.58}$$

The terms proportional to the central charge, though infinite, are not of concern so far, since corrections proportional to $c$ to the generators are to be expected. It is, however, concerning that the algebra above does not appear to close among well-defined, finite generators; in particular, the infinite sums are indicative of an inappropriate ordering. Indeed, it can be easily seen that $L_n^\lambda$ and $: L_n^{\lambda,uncorr} :$ differ by infinite terms proportional to $\lambda^2 \bar{L}_{-n}$. It is interesting to ask whether a redefinition of $: L_n^\lambda :$ can bring us to a well-defined ordering, e.g. (3.54) or (3.55), by absorbing the divergent terms proportional to $\lambda^2 \bar{L}_{-m-n}$ into a redefinition of $L_{m+n}^\lambda$. It still seems difficult though to obtain generators that satisfy a Virasoro algebra, due to the last term on the first line, which is not of the correct form. In any case, while it may be possible to recover the ordering (3.54) or (3.55) by appropriately cancelling the infinite terms above, there does not appear to exist any systematic way to do so, which could be extrapolated to higher orders in perturbation theory.

The conclusions of this analysis are that the flow equation, even keeping a slight ambiguity in the definition of the flow operator, nicely fixes the ordering and the correction terms to the left-moving generators in a correlated manner. Not only does it make sure that the resulting algebra is closed, but it also guarantees it to be Virasoro-Kac-Moody. Once the Kac-Moody generators are built, the conformal ones can be alternatively constructed using the Sugawara procedure, which yields equivalent results and shortcuts some of the technical steps.

### The right-moving generators

The right-moving generators are given (3.2) by the corresponding classical expressions upon application of an appropriate ordering procedure. As before, it is convenient to first construct the algebra of the rescaled spectral-flow-invariant generators and of the right-moving currents, $\bar{K}_m^\lambda$, and then use the Sugawara construction to produce the full right-moving generators.

The classical expression for the spectral-flow-invariant operators $\hat{\bar{\mathcal{Q}}}_m^\lambda$ is given in (2.49). Setting $t = 0$ and remembering that $\hat{\bar{L}}_m^\lambda$ correspond to the rescaled charges $R_v \hat{\bar{\mathcal{Q}}}_m$, the expansion of the general expression (2.49), up to second order in $\lambda$, in terms of the undeformed Fourier modes is

$$\hat{\bar{L}}_m^{\lambda,uncorr} = \hat{\bar{L}}_m + \frac{\lambda}{R_v}\sum_{\ell\neq 0}\frac{\ell-m}{\ell}\hat{\bar{L}}_{m+\ell}(J_\ell - \bar{J}_{-\ell}) + \lambda^2 \sum_\ell \hat{\bar{L}}_{m+\ell}\sum_{p\neq 0,\ell}\left(1 + \frac{m^2}{2p(\ell-p)} - \frac{m}{p}\right)(J_p - \bar{J}_{-p}) \times$$

$$\times (J_{\ell-p} - \bar{J}_{p-\ell}) - \frac{m\lambda^2}{2}\sum_{\ell\neq 0}\frac{1}{\ell}[(J_\ell - \bar{J}_{-\ell})\bar{J}_\ell + \bar{J}_\ell(J_\ell - \bar{J}_{-\ell})]\hat{\bar{L}}_m + \dots \tag{3.59}$$

---

[18]Note these are different from the $L_m^\lambda(:)$ operators generated by flowing with $: \mathcal{O} :$, which are given by (3.55), to which the ordering prescription (2.37) is applied.

where $1/R_v \approx 1 + \lambda w$ in this expansion, and we have written the last term in a symmetrized way that will be convenient later. Interestingly, note that except for the last term, there are no ordering issues up to this order, since $\hat{\phi}$ commutes with $\phi'$ and $\hat{\bar{L}}_m^\lambda$ to all orders in $\lambda$. The only source of ordering ambiguities stems from the terms descending from $\widetilde{\phi}_0$, which may perhaps be fixed by requiring that it remain the spectral flow generator in the full quantum theory. In any case, this is a separate issue.

The most efficient way to fix the quantum corrections to $\hat{\bar{L}}_m^\lambda$ - which are all proportional to the central charge - is by integrating the flow equation (3.9) up to $\mathcal{O}(\lambda^2)$. If we use the non-normal-ordered $\tilde{\mathcal{O}}$ to drive the flow up to $\mathcal{O}(\lambda^2)$, we obtain precisely the above expression (with the particular ordering we displayed), plus a correction proportional to

$$
\Delta\hat{\bar{L}}_m^\lambda \;=\; \frac{\lambda(c-1)}{12 R_v}(m^2-1)(J_{-m}-\bar{J}_m)_{m\neq 0} + \frac{\lambda^2(c-1)}{24}\sum_{\ell\neq 0,-m}\left[\frac{\ell-m}{\ell}((\ell+m)^2-1) - \frac{m}{\ell}(m^2-1)\right] \times
$$
$$
\times \;\; (J_\ell - \bar{J}_{-\ell})(J_{-\ell-m}-\bar{J}_{\ell+m}) \tag{3.60}
$$

Note that there is a correction already at $\mathcal{O}(\lambda)$, which contributes to the flow equation at the next order. If we flow instead using the normal-ordered $:\tilde{\mathcal{O}}:$, the only change is to replace the last term in (3.59) by its normal-ordered version.

One can check that the algebra of the corrected $\hat{\bar{L}}_m^\lambda = \hat{\bar{L}}_m^{\lambda,uncorr} + \Delta\hat{\bar{L}}_m^\lambda$ is precisely Virasoro with central extension $c-1$. We also compute the commutation relations with the left-moving generators. First, it is obvious that

$$
[\hat{\bar{L}}_m^\lambda, \hat{L}_n^\lambda] = 0 , \quad \forall \lambda \tag{3.61}
$$

to all orders in $\lambda$, due to the fact that the expansion of the right-moving generators only contains modes that commute with $\hat{L}_n = \hat{L}_n^\lambda$. The commutators with the current are however non-trivial

$$
[\hat{\bar{L}}_m^\lambda, K_n^\lambda] = \frac{\lambda k m}{2}(1+\lambda J_0)\hat{\bar{L}}_m^\lambda \delta_{n,0} \tag{3.62}
$$

and agree with the expansion of the corresponding classical expression to second order in $\lambda$. The commutator with $L_n^\lambda$ follows from the two above, via the Sugawara construction.

The expression for the right-moving $U(1)$ charges, obtained by expanding (3.2) to second order, is

$$
\bar{K}_m^{\lambda,uncorr} \;=\; \bar{J}_m + \frac{\lambda k}{2}\bar{L}_m - \frac{\lambda m}{R_v}\sum_{\ell\neq 0}\frac{1}{\ell}(J_\ell - \bar{J}_{-\ell})\left(\bar{J}_{\ell+m} + \frac{\lambda k}{2}\bar{L}_{\ell+m}\right) + \frac{\lambda^2 k}{2}\sum_\ell J_\ell \bar{L}_{m+\ell} +
$$
$$
+ \;\; \frac{m^2\lambda^2}{2}\sum_{\ell,p\neq 0}\frac{1}{\ell p}(J_\ell-\bar{J}_{-\ell})(J_p-\bar{J}_{-p})\bar{J}_{\ell+p+m} - \lambda^2 m\sum_{\ell\neq 0}\frac{1}{\ell}(J_\ell-\bar{J}_{-\ell})\bar{J}_\ell \bar{J}_m \tag{3.63}
$$

To determine the ordering and other corrections, we use the flow equation iteratively twice, taking into account the fact that it is no longer homogenous. If we use the non-normal-ordered flow operator $\tilde{\mathcal{O}}$ to fix all $\bar{K}_m^\lambda$ with $m\neq 0$, we obtain the expression above (with the terms entering the $\ell$ sums appropriately symmetrized, as e.g. in (3.59)), plus a correction

$$
\Delta\bar{K}_m^\lambda = \frac{\lambda^2 k c}{48}(m^2-1)(J_{-m}-\bar{J}_m)_{m\neq 0} \tag{3.64}
$$

The flow equation obeyed by the corrected $\bar{K}_m^\lambda \equiv \bar{K}_m^{\lambda,uncorr} + \Delta\bar{K}_m^\lambda$ is

$$
\tilde{\mathcal{D}}_\lambda \bar{K}_m^\lambda = \frac{k}{2}(\bar{L}_0 + \lambda\sum_p J_p \bar{L}_p + \lambda J_0 \bar{L}_0)\delta_{m,0} \tag{3.65}
$$

in perfect agreement with the classical result (2.34). The commutator of two such currents is

$$
[\bar{K}_m^\lambda, \bar{K}_n^\lambda] = \frac{km}{2}\delta_{m+n} + (1+\lambda J_0)\left(\frac{\lambda km}{2}\bar{K}_m^\lambda\delta_{n,0} - \frac{nk\lambda}{2}\bar{K}_n^\lambda\delta_{m,0}\right) + \mathcal{O}(\lambda^3) \tag{3.66}
$$

again in perfect agreement with the classical prediction (2.24), up to this order in perturbation theory. Next, we compute

$$[\hat{\bar{L}}_m^\lambda, \bar{K}_n^\lambda] = \frac{\lambda km}{2}(1 + \lambda J_0)\hat{\bar{L}}_m^\lambda \delta_{n,0} \tag{3.67}$$

in agreement with (2.50), with no additional corrections from the central terms. It is also trivial to show that $[\bar{K}_m^\lambda, \hat{L}_n^\lambda] = 0$ to all orders in $\lambda$, with the expected consequences for the $[\bar{K}_m^\lambda, L_n^\lambda]$ commutator. Finally, we also find

$$[\bar{K}_m^\lambda, K_n^\lambda] = \frac{\lambda km}{2}(1 + \lambda J_0)\bar{K}_m^\lambda \delta_{n,0} \tag{3.68}$$

which agrees with (2.25).

It is interesting to ask what happens if we use $: \tilde{\mathcal{O}} :$ instead to drive the flow. The correction term becomes

$$\Delta\bar{K}_m^\lambda = \frac{\lambda^2 kc}{48}(m^2 - 1)(J_{-m} - \bar{J}_m)_{m\neq 0} - \frac{\lambda^2 k}{8}m(m-1)\bar{J}_m - \frac{\lambda^2 mk}{4}(J_{-m} - \bar{J}_m)\sum_1^\infty 1 \tag{3.69}$$

in addition to normal-ordering all the terms in (3.63) that require it[19]. The last term above is problematic, because it contributes an infinite central term to the $[\bar{K}_m^\lambda, K_n^\lambda]$ commutator (the finite second term induced by the normal ordering cancels out.) We thus conclude that the normal-ordered flow operator $: \tilde{\mathcal{O}} :$, which was in principle consistent up to at least $\mathcal{O}(\lambda)$ with the defining equation (3.5), does not produce symmetry generators that have a well-defined algebra. On the other hand, the non-normal-ordered $\tilde{\mathcal{O}}$ does appear to yield well-defined results, at least up to this order.

The full unflowed generators can be built using the Sugawara construction (3.12), using the normal ordering (3.15) that preserves the usual hermiticity condition (3.18). Their commutation relations follow from those of the building blocks. It is interesting to compare $\bar{L}_m^{\lambda,S}$ with the expression that would be obtained via the flow equation. Since the analysis is rather cumbersome, we only perform the comparison up to $\mathcal{O}(\lambda)$. We find

$$\bar{L}_m^\lambda = R_v\bar{L}_m + \lambda\sum_p J_p\bar{L}_{p+m} - \frac{m\lambda}{2}\sum_{\ell\neq 0}\frac{1}{\ell}[(J_\ell - \bar{J}_{-\ell})\bar{L}_{\ell+m} + \bar{L}_{\ell+m}(J_\ell - \bar{J}_{-\ell})] + \frac{\lambda c}{12}(m^2 - 1)(J_{-m} - \bar{J}_m)_{m\neq 0} \tag{3.70}$$

This ordering is generated by flowing with the $\hat{\mathcal{O}}$ (since $\Delta\tilde{\mathcal{O}}$ vanishes at leading order), and by fixing the inhomogenous terms in the flow equation to be

$$\tilde{\mathcal{D}}_\lambda\bar{L}_m^\lambda = \frac{1}{2}(\bar{J}_m\bar{L}_0 + \bar{L}_0\bar{J}_m) \tag{3.71}$$

One can show that the above expression formally coincides with the Sugawara one, if one plugs in the explicit expressions for $\hat{L}_m^\lambda$ and $\bar{K}_m^\lambda$ to this order, through subtle cancellations of infinite terms. The match is however not perfect: in the term proportional to the central charge, we could not get the shift from $c - 1$ in (3.60) to $c$ in (3.70).

It is an interesting question which method: the flow equation or the Sugawara construction, is best for defining $\bar{L}_m^\lambda$, and whether they produce equivalent results. In our opinion, the flow equation appears more suitable, despite the fact that the exact non-homogenous terms are not known a priori, because it uniformly applies the same ordering. The Sugawara construction, on the other hand, provides a simpler way to achieve finite results that only differ from the expected expression by a finite number of terms.

We have so far mostly concentrated our attention on the unflowed generators. The flowed right-moving generators are found by solving the homogenous flow equation (3.6). In the case of $\widetilde{\bar{K}}_m^\lambda$, the result simply amounts to replacing the zero mode of $\bar{K}_m^\lambda$ by $\bar{J}_0$. Therefore, their algebra is given by (3.66), except for $m, n = 0$, when one obtains instead a trivial commutator. Consequently,

$$[\widetilde{\bar{K}}_m^\lambda, \widetilde{\bar{K}}_n^\lambda] = \frac{km}{2}\delta_{m+n} + \mathcal{O}(\lambda^3), \qquad [\hat{\bar{L}}_m^\lambda, \widetilde{\bar{K}}_n^\lambda] = 0 + \mathcal{O}(\lambda^3) \tag{3.72}$$

---

[19] The last term in (3.63) is ordered as $: (J_\ell - \bar{J}_{-\ell})\bar{J}_\ell : \bar{J}_m$.

One can in principle arrive at the same result starting from the definition (3.10), by subtracting from (3.66) the commutation relations for $H_R$. This method is significantly more cumbersome, but can be used as a cross-check. One should in particular be careful to include quantum correction terms to $H_R$ (e.g. of the form (3.60)), without which one does not obtain the correct result.

As for the flowed right-moving pseudoconformal generators, they can be constructed from $\hat{\bar{L}}_m^\lambda$ and $\widetilde{\bar{K}}_m^\lambda$ via the Sugawara procedure (3.8). Since $\hat{\bar{L}}_m^\lambda$ commutes with $\widetilde{\bar{K}}_m^\lambda$ and the former satisfy a Virasoro algebra with central charge $c - 1$, it follows that $\widetilde{\bar{L}}_m^\lambda$, $\widetilde{\bar{K}}_m^\lambda$ satisfy a Virasoro-Kac-Moody algebra with central charge $c$. Of course, we could have arrived at the same expression by solving the flow equation (3.6) for these generators, but the computation would have been significantly more cumbersome than the one presented herein.

The computations worked out in this section exemplify how our somewhat abstract definitions for the symmetry generators make very concrete predictions for their ordering and quantum corrections. Up to $\mathcal{O}(\lambda)$, our formulae for the flowed generators are similar to those worked out in [30]; however, their expressions correspond to a slightly different choice of $J\bar{T}$ flow operator, i.e. one constructed from the chiral $U(1)$ current, as opposed to the topological current we chose herein.

## 3.3. Algebra of the unflowed generators

In section 3.1, we have given a general argument that the algebra of the flowed generators consists of two commuting copies of the Virasoro-Kac-Moody algebra, with the same central extension as that of the original CFT, to all orders in $\lambda$ and $\hbar$. We have moreover argued that the following relations between the flowed and unflowed operators hold:

$$L_m^\lambda \equiv \widetilde{L}_m^\lambda + \lambda \widetilde{K}_m^\lambda H_R + \frac{k\lambda^2}{4} H_R^2 \delta_{m,0} , \qquad K_m^\lambda = \widetilde{K}_m^\lambda + \frac{\lambda k}{2} H_R \delta_{m,0}$$

$$\bar{L}_m^\lambda \equiv \widetilde{\bar{L}}_m^\lambda + \lambda \widetilde{\bar{K}}_m^\lambda H_R - \lambda \xi_m \widetilde{\bar{K}}_m^\lambda \alpha_m^r + \frac{k\lambda^2}{4} H_R^2 \delta_{m,0} , \qquad \bar{K}_m^\lambda = \widetilde{\bar{K}}_m^\lambda + \frac{\lambda k}{2} H_R \delta_{m,0} \tag{3.73}$$

The term proportional to $\xi_m$ in the right-moving pseudoconformal generator represents the different ordering prescription for this generator for different values of $m$, discussed in section 3.1. Using the explicit form of the commutators

$$[\widetilde{\bar{L}}_m^\lambda, H_R] = \widetilde{\bar{L}}_m^\lambda \alpha_m^r , \qquad [\widetilde{\bar{K}}_m^\lambda, H_R] = \widetilde{\bar{K}}_m^\lambda \alpha_m^r \tag{3.74}$$

this ordering difference can be written in the form above, where $\alpha_m^r$ is defined in (3.24). As explained, the value $\xi_m = \Theta(m)$ is the one consistent with the usual Hermiticity condition (3.18) of the generators; however, in this section we leave $\xi_m$ arbitrary, so as to capture as many ordering choices as possible.

Given the input that $\widetilde{\bar{L}}_m^\lambda$ and $\widetilde{\bar{K}}_m^\lambda$ satisfy a Virasoro-Kac-Moody algebra with central charge c to all orders, we would like to derive the algebra satisfied by the generators (3.73) to all orders in $\lambda, \hbar$. Using (3.74), the algebra of the deformed operators is

$$[\bar{K}_m^\lambda, \bar{K}_n^\lambda] = \frac{km\hbar}{2} \delta_{m+n} - \frac{\lambda k}{2} \bar{K}_n \alpha_n^r \delta_{m,0} + \frac{\lambda k}{2} \bar{K}_m \alpha_m^r \delta_{n,0} \tag{3.75}$$

$$
\begin{aligned}
[\bar{L}_m^\lambda, \bar{K}_n^\lambda] =\ & -n\hbar \bar{K}_{m+n}^\lambda - \lambda \bar{K}_m^\lambda \bar{K}_n^\lambda \alpha_n^r + \frac{\lambda k}{2} \bar{L}_m^\lambda \alpha_m^r \delta_{n,0} - \frac{\lambda^2 k}{4} \bar{K}_n^\lambda (\alpha_n^r)^2 \delta_{m,0} + \\
& + \lambda \xi_m \left[ \bar{K}_m^\lambda \bar{K}_n^\lambda (\alpha_m^r + \alpha_n^r - \alpha_{m+n}^r) - \frac{km\hbar}{2} \alpha_m^r \delta_{m+n} \right]
\end{aligned} \tag{3.76}
$$

$$
\begin{aligned}
[\bar{L}_m^\lambda, \bar{L}_n^\lambda] =\ & (m-n)\hbar \bar{L}_{m+n}^\lambda + \frac{c\hbar}{12} m(m^2 - 1) \delta_{m+n} - \lambda \bar{K}_m^\lambda \bar{L}_n^\lambda \alpha_n^r + \lambda \bar{K}_n^\lambda \bar{L}_m^\lambda \alpha_m^r - \frac{k\lambda^2}{4} \bar{L}_n^\lambda (\alpha_n^r)^2 \delta_{m,0} + \\
& + \frac{k\lambda^2}{4} \bar{L}_m^\lambda (\alpha_m^r)^2 \delta_{n,0} + \lambda (\xi_m \bar{K}_m^\lambda \bar{L}_n^\lambda - \xi_n \bar{K}_n^\lambda \bar{L}_m^\lambda)(\alpha_m^r + \alpha_n^r - \alpha_{m+n}^r) + \lambda \hbar (m-n) \xi_{m+n} \bar{K}_{m+n}^\lambda \alpha_{m+n}^r + \\
& + \lambda \hbar \bar{K}_{m+n}^\lambda (n \xi_n \alpha_n^r - m \xi_m \alpha_m^r) + \frac{\lambda^2 m \hbar k}{2} \xi_m \xi_n \alpha_m^r \alpha_n^r \delta_{m+n}
\end{aligned} \tag{3.77}
$$

It is trivial to show that the left-moving generators $L_m^\lambda, K_m^\lambda$ satisfy a Virasoro-Kac-Moody algebra with central charge $c$. The commutation relations between the left- and the right-moving generators are

$$[K_m^\lambda, \bar{K}_n^\lambda] = -\frac{\lambda k}{2}\bar{K}_n^\lambda \alpha_n^r \delta_{m,0} , \qquad [L_m^\lambda, \bar{K}_n^\lambda] = -\lambda K_m^\lambda \bar{K}_n^\lambda \alpha_n^r - \frac{\lambda^2 k}{4}\bar{K}_n^\lambda (\alpha_n^r)^2 \delta_{m,0}$$

$$[K_m^\lambda, \bar{L}_n^\lambda] = -\frac{\lambda k}{2}\bar{L}_n^\lambda \alpha_n^r \delta_{m,0} , \qquad [L_m^\lambda, \bar{L}_n^\lambda] = -\lambda K_m^\lambda \bar{L}_n^\lambda \alpha_n^r - \frac{\lambda^2 k}{4}\bar{L}_n^\lambda (\alpha_n^r)^2 \delta_{m,0} \qquad (3.78)$$

which agree with the classical commutators (2.25) at leading order in $\hbar$.

Let us now discuss a few properties of this algebra. In the appendix, it is shown that the commutators of $H_R$ with the $\widetilde{L}_m^\lambda, \widetilde{K}_m^\lambda$ operators satisfy the Jacobi identities. Since sums and products of operators satisfying the Jacobi identities also satisfy them, it follows from the definitions (3.73) that the above algebra is consistent, for any choice of the parameters $\xi_m$. Note that the potential ordering ambiguity parameterized by the latter only enters the algebra at $\mathcal{O}(\hbar^2)$, and is thus consistent with the classical results.

One may try to simplify the above expressions in the special case $\xi_m = \Theta(m)$, obtaining e.g.

$$[\bar{L}_m^\lambda, \bar{K}_n^\lambda] = -n\hbar \bar{K}_{m+n}^\lambda + \frac{\lambda k}{2}\bar{L}_m^\lambda \alpha_m^r \delta_{n,0} + \begin{cases} -\lambda \bar{K}_n^\lambda \alpha_n^r \bar{K}_m^\lambda + \frac{\lambda^2 k}{4}\bar{K}_n^\lambda (\alpha_n^r)^2 \delta_{m,0} & \text{for} \quad m \geq 0 \\ -\lambda \bar{K}_m^\lambda \bar{K}_n^\lambda \alpha_n^r - \frac{\lambda^2 k}{4}\bar{K}_n^\lambda (\alpha_n^r)^2 \delta_{m,0} & \text{for} \quad m \leq 0 \end{cases} \qquad (3.79)$$

which effectively amounts to an ordering prescription of the non-linear terms in the algebra that depends on the sign of $m$. The $[\bar{L}_m^\lambda, \bar{L}_n^\lambda]$ commutator does not appear to significantly simply, so we will omit rewriting it. It is worth noting though that the last term in (3.77) vanishes for this choice of $\xi_m$, since $m, n$ cannot be simultaneousy positive.

The algebra (3.77) can also be rewritten in terms of $\alpha_m^l$, using the identity (3.26). Remembering that $\alpha_m^r$ is better defined for large positive $m$ and $\alpha_m^l$ for large negative $m$, one can choose the best rewriting, depending on the particular signs of sign of $m, n$.

Finally, note that, unlike the Virasoro algebra, which has a finite-dimensional subalgebra generated by $L_{0,\pm 1}$, the algebra of the unflowed generators $\bar{L}_{\pm 1,0}^\lambda$ does not close; instead, they generate the entire tower of $U(1)$ generators, $\bar{K}_m^\lambda$. In particular, the only notion of $SL(2,\mathbb{R})$ Casimir is the one associated to the precise combinations of $\bar{L}_{\pm 1,0}^\lambda$ and $\bar{K}_{\pm 1,0}^\lambda$ that correspond to the flowed generators of the $SL(2,\mathbb{R})$ subalgebra.

## 3.4. Comments on representation theory

The definition of the flowed generators $\widetilde{L}_m^\lambda$, etc., via the same flow equation as that obeyed by the energy eigenstates implies that the states in $J\bar{T}$ - deformed CFTs are organised in the same way as in usual CFTs, namely in highest-weight representations of the Virasoro algebra[20]. One thus has primary states, which satisfy

$$\widetilde{L}_m^\lambda |h\rangle_\lambda = \widetilde{K}_m^\lambda |h\rangle_\lambda = \widetilde{\bar{L}}_m^\lambda |h\rangle_\lambda = \widetilde{\bar{K}}_m^\lambda |h\rangle_\lambda = 0 , \quad \forall m > 0 \qquad (3.80)$$

and

$$\widetilde{L}_0^\lambda |h\rangle_\lambda = h|h\rangle_\lambda , \qquad \widetilde{K}_0^\lambda |h\rangle_\lambda = q|h\rangle_\lambda , \qquad \widetilde{\bar{L}}_0^\lambda |h\rangle_\lambda = \bar{h}|h\rangle_\lambda , \qquad \widetilde{\bar{K}}_0^\lambda |h\rangle_\lambda = \bar{q}|h\rangle_\lambda \qquad (3.81)$$

i.e., their eigenvalues are identical to those of the corresponding states in the undeformed CFT. Descendant states are then obtained by acting with $\widetilde{L}_{-m}^\lambda$, etc, and they generate the entire Verma module. Note that the norm of these states is identical to the norm of the corresponding undeformed ones, which are real and positive, and none of the problems of $J\bar{T}$ - deformed CFTs on a cylinder, such as the existence of imaginary energies, are easily visible from this picture. It certainly appears to be a possibility that, if

---

[20]We will assume that these are the only relevant representations in the seed CFT.

$|h\rangle_\lambda$ are well defined vectors in the Hilbert space, these problems could be dissociated from the theory, to only belong to the Hamiltonian operator[21].

The primary states $|h\rangle_\lambda$ are clearly primary also with respect to the unflowed generators $\bar{L}_m^\lambda, \bar{K}_m^\lambda$.

$$L_m^\lambda |h\rangle_\lambda = K_m^\lambda |h\rangle_\lambda = \bar{L}_m^\lambda |h\rangle_\lambda = \bar{K}_m^\lambda |h\rangle_\lambda = 0 , \quad \forall m > 0 \tag{3.82}$$

which follows from their definition (3.73). One can thus equally well generate the spectrum of the $J\bar{T}$ - deformed CFT using $L_{-m}^\lambda, K_{-m}^\lambda$, etc. However, the states so generated do not have well-defined norms; for example, we have

$$_\lambda\langle h| L_m^\lambda L_{-m}^\lambda |h\rangle_\lambda = 2m\hbar\, _\lambda\langle h| L_0^\lambda |h\rangle_\lambda + \frac{c\hbar}{12}m(m^2 - 1) \tag{3.83}$$

$$_\lambda\langle h| \bar{L}_m^\lambda \bar{L}_{-m}^\lambda |h\rangle_\lambda = 2m\hbar\, _\lambda\langle h| \bar{L}_0^\lambda |h\rangle_\lambda + \frac{c\hbar}{12}m(m^2 - 1) \tag{3.84}$$

so their norm becomes imaginary at large enough initial dimension. The resulting states thus need to be treated perturbatively in $\lambda$. Again, one cannot exclude the possibility that the state vectors are well-defined, but the $L_m^\lambda$ etc., are simply not a very natural set of symmetry generators to define, in addition to being significantly more cumbersome to use than the $\widetilde{L}_m^\lambda$.

# 4. More general non-local CFTs

The argument that we have ultimately given in section 3.1 in favour of the existence of Virasoro symmetry in $J\bar{T}$ - deformed CFTs is strikingly simple and general: consider a family of non-local two-dimensional QFTs, assumed to be UV complete, which are obtained via an integrable irrelevant deformation of a CFT$_2$ (of central charge $c$), labeled by a continuous parameter $\lambda$. In order to have Virasoro symmetry, all that is required is:

i) that the deformation does not change the Hilbert space, but only the Hamiltonian and its associated eigenstates. The latter should satisfy a smooth flow equation

$$\partial_\lambda |n\rangle_\lambda = \tilde{\mathcal{O}} |n\rangle_\lambda \tag{4.1}$$

Then, one can always formally define a set of generators[22] $L_m^\lambda$ via the flow equation (3.6). By construction, the $L_m^\lambda$ will obey a Virasoro *algebra*, with central extension $c$. Note this does not by itself imply that the theory has Virasoro symmetry.

ii) if $L_0^\lambda$ is a known function of just the Hamiltonian (and, possibly, of the momentum and other conserved charges that commute with all the generators), it immediately follows that

$$[L_m^\lambda, H] = \alpha_m(H, P, Q) L_m^\lambda \tag{4.2}$$

The generators $L_{m,S}^\lambda(t) \equiv e^{i\alpha_m(H,P,Q)t} L_{m,S}^\lambda(0)$ then satisfy the conservation requirement (1.3). Therefore, the Virasoro generators defined above actually correspond to *symmetries*.

Of course, explicitly defining the non-local QFT in terms of the irrelevant deformation is a daunting task, as is proving the the resulting theory is UV complete. Since the generators $L_m^\lambda$ are defined using the flow operator $\tilde{\mathcal{O}}$, which in turn is directly related via (3.5) to the deforming operator that defines the QFT, it is equally challenging to construct the Virasoro generators explicitly. However, for many purposes, such as studying the constraints imposed by the symmetry on various observables, the explicit form of the

---

[21]To illustrate this point, consider at fixed $\lambda$ and $R$ a primary state whose energy is real. There is no reason to believe that the corresponding state vector $|h\rangle_\lambda$ is not well-defined. Consider next the descendant state obtained by acting with $\widetilde{L}_{-n}^\lambda$ on this state. There is no apparent reason that this new state vector should be ill-defined, either; yet, its energy, as measured by $H = L_0^\lambda/R + \bar{L}_0^\lambda/R_v$, is imaginary for large enough $n$. This appears to be more of a problem with the operator measuring the energy, than with the state itself.

[22]In this section, we drop the bars and the tildes from the operators, for simplicity.

generators is not needed, but only the fact that they *exist*. This follows directly from the well-definiteness of the deformation, which in the cases of interest is related to the existence of the decoupling limit.

One piece of information that one does need to know explicitly in order to establish the Virasoro generators as symmetries is the relation between $L_0^\lambda$ and the Hamiltonian of the theory, which determines the commutator of the $L_m^\lambda$ with $H$. In $J\bar{T}$ and $T\bar{T}$ - deformed CFTs, this relation can be inferred from the complete knowledge of the deformed energy spectrum, as nicely explained in [30]. Of course, the simplicity of these relations is a direct consequence of the fact that the deformed energies are given by a universal formula, which only depends on the initial energies. It is an interesting question whether similar, universal relations can be obtained in more general theories, and what is the most general relation between $L_0^\lambda$ and $H$ (and possibly other operators) that allows for the construction of an infinite set of conserved charges.

In the following, we will discuss several examples of non-local, UV complete QFTs that satisfy the conditions i) and ii) above, denoted as *non-local CFTs*. We will allow ourselves to relax condition ii) as needed, only requiring that it should be possible to add an appropriate time dependence to the Virasoro generators, so that they satisfy the conservation equation (1.3).

### $T\bar{T}$ - deformed CFTs

The conditions i) and ii) above are straightforwardly satisfied by the $T\bar{T}$ deformation. First, $T\bar{T}$ - deformed CFTs are believed to be well-defined two-dimensional QFTs obtained by specifying the irrelevant flow [15]; they were in addition given a fully non-perturbative definition in [22]. Second, their finite-size spectrum is given by

$$E_\mu(R) = \frac{\sqrt{1 + 4\mu(h + \bar{h}) + 4\mu^2(h - \bar{h})^2} - 1}{2\mu} \tag{4.3}$$

where $h, \bar{h}$ are the original conformal dimensions, $\mu$ is the $T\bar{T}$ deformation parameter, and we have set $R = 1$. The Virasoro generators, $L_m^\mu$, can then be constructed via the $T\bar{T}$ analogue of (3.6). Using the argument of [30] to extract the relation between $H, P$ and $L_0^\mu, \bar{L}_0^\mu$, the above equation implies that the zero modes of this Virasoro algebra are related to the Hamiltonian and momentum operators as

$$L_0^\mu = \frac{H + P}{2}(1 + \mu(H - P)) , \qquad \bar{L}_0^\mu = \frac{H - P}{2}(1 + \mu(H + P)) \tag{4.4}$$

These relations immediately determine the commutators of the Virasoro generators with the Hamiltonian, which take the form

$$[L_m^\mu, H] = \alpha_m(H, P)L_m^\mu , \qquad [\bar{L}_m^\mu, H] = \bar{\alpha}_m(H, P)\bar{L}_m^\mu \tag{4.5}$$

where

$$\alpha_m(H, P) = \frac{1}{2\mu}\left(\sqrt{(1 + 2\mu H)^2 + 4\mu m\hbar(1 + 2\mu P) + 4\mu^2 m^2\hbar^2} - (1 + 2\mu H)\right) \tag{4.6}$$

and $\bar{\alpha}_m$ is given by the same expression, but with $P \to -P$. The operators

$$L_{m,S}^\mu(t) \equiv e^{i\alpha_m(H,P)t} L_{m,S}^\mu(0) , \qquad \bar{L}_{m,S}^\mu(t) \equiv e^{i\bar{\alpha}_m(H,P)t} \bar{L}_{m,S}^\mu(0) \tag{4.7}$$

then have conserved expectation value in any state.

From this, we immediately conclude that $T\bar{T}$ - deformed CFTs have a Virasoro symmetry of a similar type to the one studied in detail in this article. The only qualitative difference between the two theories is that while in $J\bar{T}$ - deformed CFTs, the left-moving Virasoro symmetry is of a standard type, i.e. implemented by local conformal transformations, in the $T\bar{T}$ case both Virasoro symmetries will be implemented by fully non-local transformations. This is directly related to the fact that the $J\bar{T}$ deformation produces a 'half - nonlocal' theory (more precisely, a dipole CFT), whereas the $T\bar{T}$ deformation produces a fully non-local one.

While the quantum-mechanical arguments we have just provided apply equally well to $T\bar{T}$ - deformed CFTs as they do to $J\bar{T}$ - deformed ones, in the latter case we have additionally been able to provide a

completely explicit, fully non-linear classical realisation of the conserved charges (as well as a perturbative quantum construction). By contrast, in classical $T\bar{T}$ - deformed CFTs, Virasoro symmetries have currently been shown to exist in the non-compact case only, when the subtleties associated with boundary conditions can be ignored[23]. It would be certainly interesting to find the classical conserved charges for $T\bar{T}$ - deformed CFTs in compact space, e.g. by following the steps outlined in [25] for the case of $J\bar{T}$.

### Single-trace $T\bar{T}$ and $J\bar{T}$ - deformed CFTs

Another set of theories that are likely to satisfy the two conditions we spelled out, appropriately relaxed, correspond to deformations of a symmetric product orbifold CFT by a "single-trace" $T\bar{T}$ and $J\bar{T}$ deformation, i.e. an operator of the form

$$\sum_{i=1}^{N} T_i \bar{T}_i \qquad \text{or} \qquad \sum_{i=1}^{N} J_i \bar{T}_i \tag{4.8}$$

where the index $i$ runs over the $N$ copies of the CFT. For specific choices of the seed CFT, such theories have been proposed to be holographically dual to the background obtained from the NS5 decoupling limit of the NS5-F1 system [32] and, respectively, to a decoupling limit that yields a warped AdS$_3$ spacetime [16, 23].

We concentrate on the single-trace $T\bar{T}$ deformation (the $J\bar{T}$ case is entirely analogous). We will moreover focus on the untwisted sector only, where the effect of the single-trace $T\bar{T}$ deformation on the spectrum is clearly understood, as it acts independently in each copy. In this case, one can build a Virasoro algebra $L_m^{\mu,i}$ associated to each of the copies, and expect that the corresponding $L_0^{\mu,i}$ is related via (4.4) to the Hamiltonian $H_i$ associated to that copy. The total Hamiltonian can then be written as

$$H = \sum_{i=1}^{N} \frac{\sqrt{1 + 4\mu\left(L_0^{\mu,i} + \bar{L}_0^{\mu,i}\right) + 4\mu^2\left(L_0^{\mu,i} - \bar{L}_0^{\mu,i}\right)^2} - 1}{2\mu} \tag{4.9}$$

The total Virasoro generators can be built as $L_m^\mu = \sum_i L_m^{\mu,i}$ and are manifestly symmetric under the interchange of the copies. Since the various copies commute with each other, we find

$$[L_m^\mu, H] = \sum_i \alpha_m(H^i, P^i) L_m^{\mu,i} \tag{4.10}$$

Given this commutator, it is straightforward to construct a generator whose eigenvalue is conserved

$$L_{m,S}^\mu(t) = \sum_i e^{i\alpha_m(H^i, P_i)t} L_{m,S}^{\mu,i} = \sum_i L_{m,S}^{\mu,i}(t) \tag{4.11}$$

Since the states in the symmetric product orbifold are symmetrized, the expectation values of $H_i$ across the copies should be equal, thus producing an overall time-dependent phase factor, which would be interesting to reproduce from a gravitational calculation. For such a symmetric state, in which $\langle H_i \rangle = \langle H \rangle / N$, it is interesting to note that the departure of $\alpha_m$ from its usual CFT value is

$$\langle \alpha_m \rangle = m\hbar - \frac{2m\mu\hbar}{N} \frac{\langle H \rangle - \langle P \rangle}{1 + 2\mu m\hbar} + \mathcal{O}(1/N^2) \tag{4.12}$$

suggesting that even if the non-locality scale of the theory is set by $\mu$, the departure of the non-local Virasoro algebra from a standard one, as measured by the commutator $\alpha_m$, is suppressed by powers of $1/N$, and thus possibly hard to detect in the dual gravity picture. If the corrections associated to the inclusion of twisted sectors do not significantly modify this picture, this 'large N' supression mechanism of the non-local commutator could provide an explanation for the ubiquity of Virasoro asymptotic symmetry groups in three-dimensional spacetimes that are not asymptotically AdS$_3$.

---

[23]The conserved charges studied in [24] are only well defined in infinite volume. In finite volume, the field-dependent coordinates proposed therein do not satisfy all possible Jacobi identities [31]. An encouraging sign coming from the study of $J\bar{T}$ - deformed CFTs suggests that the zero mode of the field-dependent coordinate, which is the one that prevents the Jacobi identities from being satisfied in $T\bar{T}$, may in fact drop out from the expression for the improved finite-size charges.

*More general deformations*

As noted in the introduction, there exist many examples of string backgrounds that are obtained through non-trivial decoupling limits, and which could be dual to non-local, UV complete two-dimensional QFTs, which appear to fulfill at least the first condition on having a Virasoro algebra.

To understand whether a direct relation between $L_0^\lambda$ and $H$ can be established, as required by our second condition, one can try to analyse the spectrum of the deformed theory (representing the $H$ eigenvalues) as a function of the undeformed one (representing the eigenvalues of $L_0^\lambda$). In $T\bar{T}$ and $J\bar{T}$ - deformed CFTs, the two are related in a universal manner that does not depend at all on the details (or particular spectrum) of the seed CFT. For this reason, the universal relation between the deformed and undeformed spectra in these theories can be lifted to a general relation between the operators $H$ and $L_0^\lambda$.

For the more general deformations we are interested in, the situation is more nuanced[24]. On the one hand, reducing the existence of the Virasoro symmetry to a question about the spectrum of the theory is an enormous simplification, because the spectrum is an observable that can be computed not just in the field theory - where tracking the irrelevant deformation is hard, if not impossible - but also in the dual supergravity or string theory picture, where the description of the deformation may significantly simplify[25]. On the other hand, it is harder to argue that knowledge of the deformed spectrum will immediately imply a relation between $H$ and $L_0^\lambda$, because: i) the deformations are less universal, and therefore they will only exist in specific seed CFTs, whose spectra may not be generic and ii) even if one succeeds in computing the spectrum, this will usually be for a subsector of the theory (e.g., described by supergravity excitations, or string solutions), and thus one will not in general have access to the relation between $L_0^\lambda$ and $H$ for all possible eigenvalues that they can take.

This being said, it does appear to be true that in many string-theoretical examples of continuous deformations, either exactly marginal [34], or of the more general kind that change the asymptotics [35], the deformed dimensions are (especially at strong coupling) smooth, universal functions of the undeformed ones for entire subsectors of the theory. One may therefore be able to write at least an approximate relation between $H$ and $L_0^\lambda$ in these subsectors, which may allow one to show, via an argument that parallels the one given for the $J\bar{T}$ and $T\bar{T}$ deformations, that the respective subsector is acted upon by a Virasoro symmetry. Such an aproximate relation could be sufficient to explain why various asymptotic symmetry group analyses in the dual gravity picture were uncovering Virasoro symmetries. Note the latter may even act locally, perhaps through a 'large N' supression mechanism of the non-local part of the commutator with $H$, similar to the one we hinted at in the case of single-trace $T\bar{T}$ deformations. Whether the full theory has Virasoro symmetry would however depend on the exact expression for $L_0^\lambda$ in terms of $H$ and other operators in the theory, which would presumably need to be rather special, and likely difficult to establish.

# 5. Discussion

In this article, we have analysed the symmetries of $J\bar{T}$ - deformed CFTs, at both the classical and the quantum level and showed that in a certain basis, they consist of two commuting copies of the Virasoro-Kac-Moody algebra, with the same central extension as that of the undeformed CFT. While the possibility of having Virasoro symmetries in a non-local theory may seem surprising, we provided a concrete mechanism for reconciling the non-locality of the model with these symmetries: the zero mode of the Virasoro algebra is not identified with the Hamiltonian, as in usual CFTs; rather, it is a non-linear function of it. We showed such a condition was still compatible with the existence of an infinite set of Virasoro conserved charges and discussed possible extensions of this mechanism to more general non-local, UV complete QFTs.

There are many interesting future directions to explore. First, our construction of the symmetry generators is somewhat abstract, even in the classical case, where we did provide entirely explicit formulae for the conserved charges. It would be worth having a better physical picture of the action of these non-local symmetries on the fields, as well as a better understanding of the physical implications of an

---

[24]We thank Gregory Korchemsky for discussions of this point.

[25]For example, in the case of TsT transformations, their field-theoretical description is usually intractable, whereas their effect in the dual string theory description is to simply change the boundary conditions of the worldsheet fields [33].

operator-dependent time dependence in the generators. One may be able to address these questions by focussing on a simple example, such as the $J\bar{T}$ - deformed free boson.

Still on the technical side, it would be interesting to develop an algorithmic procedure for explicitly computing quantum corrections to the $J\bar{T}$ operator, to all orders in the coupling. As explained in the main text, this would involve reconstructing the right-moving translation current $T_{\alpha V}$ from knowledge of the right-moving conserved charges. While this exercise is clearly possible in principle, it is less clear how to do it in practice, due to the non-linear nature of the charges. Perhaps reconstructing the right-moving spectral-flow-invariant current, whose relation to the associated conserved charges is simpler (2.49), may be a good place to start.

Another interesting direction would be to understand the Virasoro symmetries discussed in this article from a dual gravitational point of view. The holographic description of $J\bar{T}$ - deformed CFTs is given by AdS$_3$ gravity coupled to a Chern-Simons gauge field, with mixed boundary conditions connecting the two [36]. While the field-dependent Virasoro symmetries of $J\bar{T}$ - deformed CFTs were first uncovered in the asymptotic symmetry group analysis of these boundary conditions, that analysis did not take into account the subtleties associated with finite size which, as we have recently learned, play an essential role in rendering the charges well-defined in compact space. It would thus be very interesting to understand how the fully non-local piece of the transformation is implemented in the gravitational description, and in particular whether the current formalisms for computing asymptotic conserved charges already incorporate such structures, or not. Since field-dependent asymptotic symmetries are not uncommon in space-times with non-trivial asymptotics, this analysis could be rather useful in future applications of asymptotic symmetry group methods to non-AdS holography.

So far, we have only described the action of the non-local Virasoro generators on the spectrum of the theory. An obvious next step is to study the consequences of these symmetries on other observables, such as correlation functions. For this, one should find a basis of operators that transform nicely under them, analogously to primary operators in usual CFTs. Should one be able to fix the form of correlation functions using symmetry arguments alone, one may attempt to give an axiomatic definition of non-local CFTs that does not specifically refer to their construction in terms of an irrelevant deformation, which does not appear very natural from a renormalisation group point of view.

All the questions that we listed above also apply to the $T\bar{T}$ deformation. In this case, it would be interesting to first achieve an explicit classical realisation of the symmetry generators, as we already have for $J\bar{T}$ - deformed CFTs. One can also ask whether an analogue of the unflowed generators exists in these theories, given that the quantum argument we provided only predicts the existence of the analogues of the flowed ones. After finding the classical symmetries, it would be interesting to understand how they match to the asymptotic symmetry group analysis in the dual AdS$_3$ with mixed boundary conditions [37].

Finally, a rather interesting avenue to explore is to find more general examples of non-local CFTs, along the lines of our discussion in section 4. Possibly the simplest such generalizations are the single-trace analogues of the $T\bar{T}$ and $J\bar{T}$ deformations that we have already mentioned, whose advantage is to have a rather concrete definition. In this setting, it appears worthwhile to better understand the effects of these single trace deformations on the twisted sectors. By doing so, one would not only be able to test the holographic duality proposed in [32] at a more refined level, but also make new field-theoretical predictions that should be reproduced by the gravitational side of the correspondence. One can also consider deformations of this duality and understand what is the most general structure of the deformed CFT that still allows for the existence of a Virasoro symmetry, as well as whether, in the holographic duals to the string backgrounds discussed in the introduction, this symmetry is exact or, rather, emergent in the large $N$, large gap limit.

**Acknowledgements**

The author would like to thank Zohar Komargodsky, Gregory Korchemsky, Mark Mezei, Nikita Nekrasov, Anton Pribytok, Sylvain Ribault, Samson Shatashvili and especially Blagoje Oblak for interesting discussions. This research was supported in part by the ERC Starting Grant 679278 Emergent-BH.

# A. Properties of $\alpha_m^{l,r}$

The quantities $\alpha_m^{l,r}$ are operator-valued functions, defined via the commutation relations of the flowed generators with the right-moving Hamiltonian

$$[\widetilde{L}_m^\lambda, H_R] = \widetilde{L}_m^\lambda \alpha_m^r = \alpha_m^l \widetilde{L}_m^\lambda \tag{A.1}$$

The $\alpha_m^{l,r}$ need not commute with $\widetilde{L}_n^\lambda$, reason for which they can in principle be different. They do commute with $H_R$, being a function of it. Their explicit form to all orders in $\hbar$ can be found using (A.1) and the following relations

$$[\widetilde{L}_m^\lambda, \widetilde{L}_0^\lambda] = m\hbar\widetilde{L}_m^\lambda, \qquad \widetilde{L}_0^\lambda = R_v H_R - \lambda H_R \bar{J}_0 - \frac{k\lambda^2}{4} H_R^2 \tag{A.2}$$

both of which are assumed to be exact in $\hbar$. Commuting the second equation with $\widetilde{L}_m^\lambda$, we find

$$\frac{k\lambda^2}{4}(\alpha_m^r)^2 + (R - \lambda Q_K)\alpha_m^r - m\hbar = 0 \tag{A.3}$$

$$\frac{k\lambda^2}{4}(\alpha_m^l)^2 - (R - \lambda Q_K)\alpha_m^l + m\hbar = 0 \tag{A.4}$$

where, as before, $Q_K = J_0 + \frac{\lambda k}{2} H_R$. The solutions are given by

$$\alpha_m^r = -2\frac{R - \lambda Q_K - \sqrt{(R - \lambda Q_K)^2 + \hbar km\lambda^2}}{k\lambda^2} \approx \frac{m\hbar}{R - \lambda Q_K} - \frac{km^2\lambda^2\hbar^2}{4(R - \lambda Q_K)^3} + \mathcal{O}(\hbar^3) \tag{A.5}$$

$$\alpha_m^l(E_R) = 2\frac{R - \lambda Q_K - \sqrt{(R - \lambda Q_K)^2 - \hbar km\lambda^2}}{k\lambda^2} \approx \frac{m\hbar}{R - \lambda Q_K} + \frac{km^2\lambda^2\hbar^2}{4(R - \lambda Q_K)^3} + \mathcal{O}(\hbar^3) \tag{A.6}$$

This reproduces the classical result (2.27) to leading order in $\hbar$. It may be sometimes useful to rewrite this using $\widetilde{L}_0^\lambda$ instead of $Q_K$, using the identity

$$(R - \lambda Q_K)^2 = (R - \lambda J_0)^2 - \lambda^2 k \widetilde{L}_0^\lambda \tag{A.7}$$

Note that, to the extent that $Q_K$ has real eigenvalues, $\alpha_m^r$ is well defined for arbitrarily large positive $m$, while $\alpha_m^l$ is well defined for large negative $m$. One can consequently choose the best way to write (A.1), dpending on the value of $m$.

Using the fact that the commutator $[\widetilde{K}_m^\lambda, \widetilde{L}_0^\lambda] = m\hbar\widetilde{K}_m^\lambda$, one can also show that

$$[\widetilde{K}_m^\lambda, H_R] = \widetilde{K}_m^\lambda \alpha_m^r = \alpha_m^l \widetilde{K}_m^\lambda \tag{A.8}$$

for the same values of $\alpha_m^{l,r}$.

It is interesting to work out the commutator of $\widetilde{L}_m^\lambda$ with an arbitrary function of $H_R$. We find

$$[\widetilde{L}_m, f(H_R)] = \widetilde{L}_m(f(H_R) - f(H_R - \alpha_m^r)) = (f(H_R + \alpha_m^l) - f(H_R))\widetilde{L}_m \tag{A.9}$$

and similarly for $\widetilde{K}_m^\lambda$. One can use this to show in particular that the difference between $\alpha_m^l$ and $\alpha_m^r$ is consistent with (A.1), because

$$\alpha_n^r(H_R) = \alpha_n^l(H_R - \alpha_n^r), \qquad \alpha_n^l(H_R) = \alpha_n^r(H_R + \alpha_n^l) \tag{A.10}$$

Other useful relations are

$$\alpha_n^r(H_R - \alpha_m^r) = \alpha_{m+n}^r(H_R) - \alpha_m^r(H_R), \qquad \alpha_n^l(H_R + \alpha_m^l) = \alpha_{m+n}^l - \alpha_m^l \tag{A.11}$$

which can be used to show that

$$[\widetilde{\bar{L}}_m^\lambda, \alpha_n^r] = \widetilde{\bar{L}}_m^\lambda(\alpha_m^r + \alpha_n^r - \alpha_{m+n}^r) , \qquad [\widetilde{\bar{L}}_m^\lambda, \alpha_n^l] = (\alpha_{m+n}^l - \alpha_m^l - \alpha_n^l)\widetilde{\bar{L}}_m^\lambda \tag{A.12}$$

and similarly for $\widetilde{\bar{K}}_m^\lambda$. These relations, together with $\alpha_0^{l,r} = 0$, can be used to show that any combination of the flowed right-moving generators and $H_R$ satisfies the the Jacobi identities.

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
