# Peer review of "$J\bar T$-deformed CFTs as non-local CFTs"

_SciPost Physics_

## Round 1 · Referee Report · Anonymous (Referee 1) · 2022-3-31

Strengths

1-very well written
2-interesting topic
3-important results

Report

This article is about the interpretation of JTbar deformations of conformal field theories as non-local conformal field theories.
The primary objective is to study classical and quantum symmetry generators.

The article is well written. The background information presented in the introduction and throughout the paper is valuable to follow the logical flow and the (sometimes quite technical) calculations, which are anyway always carefully explained.
The article is self-contained.
The most relevant outcomes concern the derivation of the non-linear modification of the Witt-Kac-Moody algebra arising after the perturbation and the introduction of alternative generators fulfilling the unperturbed algebra.

The article ends with a discussion on possible generalizations to other TTbar-type perturbations.

The paper is interesting and contains many original comments and findings. I recommend its publication without hesitation.

---

## Round 1 · Referee Report · Anonymous (Referee 2) · 2022-4-6

Strengths

  • interesting new result on some particular solvable deformations
  • well-written paper

Weaknesses

  • somewhat narrow in scope
  • somewhat technical in nature

Report

The article presents some new results on the symmetries in a particular class of exactly-solvable deformations (T-Jbar theories, on which the author has worked extensively).

The main claim is that, despite the deformation, it is possible to construct a Virasoro-Kac-Moody symmetry algebra, though its relation with the energy is now non-linear. To this end, the author first analyzes the classical theory and its symmetry, and then promotes the resulting algebra generators to quantum operators.

My main confusion about this paper is that a large part of the motivation is to construct "non-local CFTs" by means of the deformation. However, the author never discusses in which sense these theories are non-local.

I understand that, when starting from a theory with a local Lagrangian, the T-Jbar deformation yields a theory with a non-polynomial Lagrangian, which is therefore non-local in the fields. However, especially in the context of CFTs in D=2, it is not uncommon to define the theory without any reference to a Lagrangian. Some of the D=2 CFTs which are known in the literature can be said to be non-local in some sense or another. The most radical (and genuine) way for a CFT to be nonlocal would be to exhibit monodromies in a correlation function (meaning: taking one operator $O(x)$ around another one $O'(0)$ would change a correlation function).

Also in light of the claim in the abstract
"We argue that J-Tbar- deformed CFTs are the first concrete realisation of such “non-local CFTs”"
I would like the author to clarify exactly in which way these models are non-local and how this is unique with respect to other known classes of CFTs in D=2.

Requested changes

1) Clarify how the models studied are non-local 2) Explain why these models are unique or in any case "the first concrete realisation" of such non-locality

---

## Round 1 · Referee Report · Anonymous (Referee 3) · 2022-4-20

Strengths

1- Explicits the procedure to express nonlocal symmetries that were expected (but not determined) in previous works on JTbar deformed CFTs.

2- Interprets these nonlocal symmetries as being relevant to holography.

Weaknesses

1- Notations for the numerous types of charges are hard to follow.

2- Definition of time-dependent symmetries is too general as stated and allows for arbitrary functions on phase space.

Report

I apologize to the author and the editor for the delay.

Motivated by holographic setups whose boundary theory appears to have nonlocal conformal symmetry, the author observes that JTbar (or TTbar) deformed 2d CFTs have such nonlocal conformal symmetry. This motivates a detailed investigation of how symmetries are deformed under JTbar. Overall the paper is nicely written, and provides an interesting holographic viewpoint on known symmetries.

After a classical analysis where the algebra of Poisson brackets of these symmetries is determined, the author embarks in the explicit construction of nonlocal quantum conserved charges, and their Lie algebra, to low order. It is not clear to me how much this part differs from the analysis of [30] (Le Floch-Mezei) which had also constructed them order by order in the coupling constant. One small difference is that the author packages all of the dependence in a single nonlocal automorphism $\tilde{\mathcal{O}}$, but the underlying calculation does not seem simplified.

An important worry I have is whether these nonlocal operators can really be understood as symmetries. The key is to understand what time dependence one should allow in a conserved charge. With the definitions in the introduction, one could call $Q_t := \exp(t\{H,\})Q = Q + t \{H,Q\} + t^2 ...$ a conserved charge for an arbitrary function Q on phase space (classically), or similarly with Lie brackets quantum mechanically. While it is true that the value of such a Q_t would be independent of time trivially, this is not a useful notion of symmetry. In the author's application to JTbar, the needed time dependence is less general and seems to only consist in an operator-valued phase factor, which seems acceptable. Likewise, boosts are ok because the conserved charge is a reasonably simple $M^{0i} + t P^i$. I guess one consequence of this paper is that I am now confused about what exactly a symmetry is!

Requested changes

1- Throughout the paper, "posses" → "possesses" (or "possess" in plural form).

2- In "a transformation of the fields and, eventually, the coordinates" change "eventually" to "possibly".

3- As stated in the report, it would be good if the author could clarify what time dependence should be allowed around footnote 2 (above equation (1.2)).

4- It seems "off-shell" below (1.2) is not the relevant point, but rather it should be said that $\partial_t Q$ is the derivative with respect to explicit time dependence. Then there is nowhere to apply equations of motion anyways, so no notion of on-shell versus off-shell.

5- Scrödinger is missing an h.

6- The notation for conserved charges is impossible to follow, for instance (2.7) defines $Q_f$ and $P_\eta$ as the modes of the Hamiltonian and U(1) symmetry. The zero mode of $P_\eta$ is $Q_k$. Later $\mathcal{P}$ is the momentum, no relation to the U(1) symmetry. It seems more practical to decide once and for all which one of P and Q corresponds to U(1) versus conformal symmetry. (My preference would be Q for U(1) and P for conformal, but that would be a pretty large change.)

7- A bunch of $Q_K$ should be $Q_k$ (lowercase), or the opposite, I'm not sure if $K$ stands for the current $K$ or for the level $k$.

8- In "due to the additional commutator ${v_{imp},\phi}$" it should be "${\Delta v,\phi}$" I think.

9- I don't understand how $\chi$ is defined below (2.29), how can it be made periodic in $\sigma$?

10- Above and in (2.31) it might be cleaner to say $X=\tilde{Q}_m,....$ satisfy

$$\tilde{D}_\lambda X=...=0.$$
Similar for other places in the paper.

11- By that point I was already struggling to keep track of notation. I advise the author to include a table of notation at the start of the paper, perhaps after the introduction. Seeing both the classical and quantum objects, flowed and unflowed, next to each other would help understand what is being computed.

12- Regarding "relax condition ii) as needed, only requiring that it should be possible to add an appropriate time dependence", I have the same objection as in my requested change 3, namely it is not clear what time dependence should be allowed to qualify as a symmetry.

---

## Editorial Decision

awaiting_resubmission